# GENERATIVE MODELING ON MANIFOLDS THROUGH MIXTURE OF RIEMANNIAN DIFFUSION PROCESSES

## ABSTRACT

Learning the distribution of data on Riemannian manifolds is crucial for modeling data from non-Euclidean space, which is required by many applications from diverse scientific fields. Yet, existing generative models on manifolds suffer from expensive divergence computation or rely on approximations of heat kernel. These limitations restrict their applicability to simple geometries and hinder scalability to high dimensions. In this work, we introduce the Riemannian Diffusion Mixture, a principled framework for building a generative process on manifolds as a mixture of endpoint-conditioned diffusion processes instead of relying on the denoising approach of previous diffusion models, for which the generative process is characterized by its drift guiding toward the most probable endpoint with respect to the geometry of the manifold. We further propose a simple yet efficient training objective for learning the mixture process, that is readily applicable to general manifolds. Our method outperforms previous generative models on various manifolds while scaling to high dimensions and requires a dramatically reduced number of in-training simulation steps for general manifolds.

## 1 INTRODUCTION

Deep generative models have shown great success in learning the distributions of the data represented in Euclidean space, e.g., images and text. While the focus of the previous works has been biased toward data in the Euclidean space, modeling the distribution of the data that naturally resides in the Riemannian manifold with specific geometry has been underexplored, while they are required for wide application: For example, the earth and climate science data (Karpatne et al., 2018; Mathieu & Nickel, 2020) lives in the sphere, whereas the protein structures (Jumper et al., 2021; Watson et al., 2022) and the robotic movements (Simeonov et al., 2022) are best represented by the group SE(3). Moreover, 3D computer graphics shapes (Hoppe et al., 1992) can be identified as a general closed manifold. However, previous generative methods are ill-suited for modeling these data as they do not take into consideration the specific geometry describing the data space and may assign a non-zero probability to regions outside the desired space.

Recent works (Bortoli et al., 2022; Huang et al., 2022) extend the diffusion generative framework to the Riemannian manifolds that learn to reverse the noising process, similar to the Euclidean diffusion models. Although diffusion models have been shown to successfully model the distribution on simple manifolds, e.g., sphere and torus, the denoising diffusion based on the time-reversal causes difficulty in training since the score matching objective either requires an approximation of the intractable heat kernel that obstructs exact training or needs computation of the divergence which is computationally expensive and scales poorly to high dimensions. In addition, previous diffusion models on the manifold are geometrically not intuitive as their generative processes are derived from the time reversal of the noising process that accompanies score function in the drift which does not provide explicit geometrical interpretation.

On the other hand, continuous normalizing flow (CNF) models on the manifold (Mathieu & Nickel, 2020; Rozen et al., 2021; Ben-Hamu et al., 2022; Chen & Lipman, 2023) aim to learn the continuous-time flow by parameterizing the vector field. Compared with the diffusion models that rely on stochastic processes, CNF models build a deterministic process and do not suffer from the intractability of the Brownian motion. However, most of the CNF models require computing divergence during training that cannot even scale to moderately high dimensions and moreover cannot be readily

adapted to general geometries. Even though few works (Rozen et al., 2021; Ben-Hamu et al., 2022; Chen & Lipman, 2023) proposed simulation-free methods on simple manifolds, they still require in-training simulation for general manifolds, and we observe that simulating the deterministic process necessitates a large number of simulation steps to obtain accurate trajectories.

To this end, we present Riemannian Diffusion Mixture, a new generative model for building a diffusion process on general Riemannian manifolds that transports an arbitrary prior distribution to the data distribution. We build upon the diffusion mixture framework (Peluchetti, 2021; Liu et al., 2023) defined in the Euclidean space, constructing a generative process directly on the manifold as a mixture of bridge processes, i.e., diffusion process conditioned to endpoints, that does not rely on the time-reversal of the noising process. Specifically, we introduce new families of tractable bridge processes on general manifolds that do not require an approximation of the heat kernels. We construct a mixture process from these bridge processes that models a stochastic map on the manifold from an arbitrary prior distribution to the data distribution, which can be characterized by its drift that guides the process toward the direction of the most probable endpoint considering the geometry of the manifold. Further, we derive an efficient training objective, namely the two-way bridge matching that can scale to high dimensions and is readily applicable to general manifolds. We establish a theoretical background for the diffusion mixture framework on the Riemannian setting and derive that the previous CNF model is a special case of our framework.

We experimentally validate our approach on diverse manifolds on both real-world and synthetic datasets, on which our method outperforms or is on par with the state-of-the-art baselines. We demonstrate that ours can scale to high dimensions while allowing significantly faster training compared to score matching of previous diffusion models. Especially on general manifolds, our method shows superior performance with dramatically reduced in-training simulation steps, using only 5% of the simulation steps compared to the previous CNF model. We summarize our main contributions as follows:

- We propose a new generative model for directly building a generative process on general manifolds, which does not rely on the time-reversal approach used in previous denoising diffusion models.
- We derive theoretical groundwork for the bridge processes and the diffusion mixture process in the Riemannian setting, and introduce an efficient objective readily applicable to general manifolds.
- Our method shows superior performance on diverse manifolds, and further can scale to higher dimensions while allowing significantly faster training compared to previous diffusion models on manifolds with known geodesics.
- Especially on general manifolds, our approach outperforms previous CNF model with greatly reduced in-training simulation steps, demonstrating the necessity of a stochastic generative process.

## 2 BACKGROUND

In this section, we introduce the basic concepts of Riemannian manifolds and briefly explain the diffusion processes defined on the manifold.

**Riemannian Manifold** We consider complete, orientable, connected, and boundaryless Riemannian manifolds $\mathcal{M}$ equipped with Riemannian metric $g$ that defines the inner product of tangent vectors. $T_x\mathcal{M}$ denotes the tangent space at point $x \in \mathcal{M}$ and $\|\eta\|_\mathcal{M}$ denotes the norm of the tangent vector $\eta \in T_x\mathcal{M}$. For smooth function $f : \mathcal{M} \to \mathbb{R}$, $\nabla$ denotes the Riemannian gradient where $\nabla f(x) \in T_x\mathcal{M}$, $\mathrm{div}(v)$ denotes the Riemannian divergence for the smooth vector field $v : T_X\mathcal{M} \to \mathcal{M}$, and $\Delta_\mathcal{M}$ denotes the Laplace-Beltrami operator defined by $\Delta f = \mathrm{div}(\nabla f)$. $\exp_x : T_x\mathcal{M} \to \mathcal{M}$ and $\exp_x^{-1} : \mathcal{M} \to T_x\mathcal{M}$ denotes the Riemannian exponential and logarithmic map, respectively. Lastly, $\mathrm{dvol}_x$ denotes the volume form on the manifold, and $\int f(x)\mathrm{dvol}_x$ denotes the integration of the function $f$ over the manifold.

**Diffusion Process on Riemannian Manifold** Brownian motion on a Riemannian Manifold $\mathcal{M}$ is a diffusion process generated by $\Delta_\mathcal{M}/2$ (Hsu, 2002) which is a generalization of the Euclidean Brownian motion. The transition distribution of the Brownian motion corresponds to the heat kernel, i.e. the fundamental solution to the heat equation, which coincides with the Gaussian distribution when $\mathcal{M}$ is a Euclidean space. One can construct a diffusion process that converges to a stationary

distribution described by the Langevin dynamics with respect to the Brownian motion $\mathbf{B}_t^{\mathcal{M}}$:

$$\mathrm{d}\boldsymbol{X}_t = -\frac{1}{2}\nabla_{\boldsymbol{X}_t} U(\boldsymbol{X}_t)\mathrm{d}t + \mathrm{d}\mathbf{B}_t^{\mathcal{M}}, \tag{1}$$

where the terminal distribution satisfies $\mathrm{d}p(x)/\mathrm{dvol}_x \propto e^{-U(x)}$ (Durmus, 2016) for a potential function $U$ which we describe in detail in Appendix A.1. A diffusion process on the manifold can be simulated using the Geodesic Random Walk (Jørgensen, 1975; Bortoli et al., 2022) which corresponds to taking a small step on the tangent space in the direction of the drift.

## 3 RIEMANNIAN DIFFUSION MIXTURE

We now present Riemannian Diffusion Mixture, a generative framework for building a diffusion process on general manifolds as a mixture of endpoint-conditioned diffusion processes.

### 3.1 BRIDGE PROCESSES ON MANIFOLD

The first step of constructing the generative process is designing a diffusion process conditioned to fixed endpoints, i.e. the bridge process. In contrast to the Euclidean space which is equipped with simple families of bridge processes derived from the Brownian motion or the Ornstein-Uhlenbeck process, designing a bridge process on general manifolds is challenging since the transition density of the Brownian motion is intractable in general. In order to achieve a simple bridge process that could be used for constructing a generative model, we start with the Brownian bridge on a compact manifold $\mathcal{M}$ with fixed starting and end points, modeled by the following SDE (see Appendix A.1 for details of the Brownian bridge):

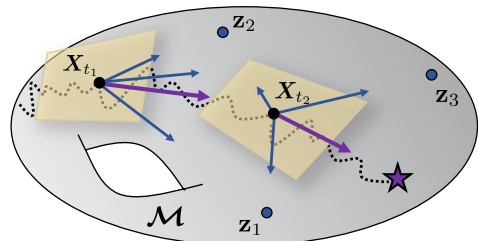

Figure 1: We construct a generative process on the manifolds as a mixture of bridge processes where its drift (**purple**) corresponds to the weighted mean of the tangent vectors pointing to the directions of the endpoints (**blue**), that guides the diffusion process (black dotted) to the most likely endpoint.

$$\mathrm{d}\boldsymbol{X}_t = \nabla_{\boldsymbol{X}_t}\log p_{\mathcal{M}}(\boldsymbol{X}_t, z, T-t)\mathrm{d}t + \mathrm{d}\mathbf{B}_t^{\mathcal{M}}, \quad \boldsymbol{X}_0 = z_0 \tag{2}$$

where $p_{\mathcal{M}}$ denotes the heat kernel on $\mathcal{M}$ and $z$ denotes the fixed endpoint. We cannot directly use this Brownian bridge process as the heat kernel is known in very limited cases and even on a simple manifold such as a sphere, the heat kernel is represented as an infinite sum (Tulovsky & Papiez, 2001). Thereby we explore a new family of bridge processes that do not require the heat kernel.

Intuitively, a diffusion process that takes each step in the direction of the endpoint should carry the process toward the desired endpoint regardless of the prior distribution. The natural choice for this direction would be following the shortest path between the current state and the endpoint on the manifold, which corresponds to the inverse of the exponential map[1], i.e., the logarithm map. The logarithm map provides a simple approach to represent a tangent vector that heads toward the desired endpoint, as illustrated in Figure 1 by the blue vectors pointing to the endpoints.

From this observation, we introduce a new family of bridge processes on manifolds for which the drift is derived from the logarithm map, namely the *Logarithm Bridge Process*[2]:

$$\mathbb{Q}_{log}^z \; : \; \mathrm{d}\boldsymbol{X}_t = \frac{\sigma_t^2}{\tau_T - \tau_t}\exp_{\boldsymbol{X}_t}^{-1}(z)\mathrm{d}t + \sigma_t\mathrm{d}\mathbf{B}_t^{\mathcal{M}}, \quad \boldsymbol{X}_0 \sim \Gamma \; ; \; \tau(t) \coloneqq \int_0^t \sigma_s^2 \mathrm{d}s. \tag{3}$$

where $z$ is the fixed endpoint, $\exp^{-1}$ is the logarithm map, $\Gamma$ denotes the prior distribution, $\sigma_t$ is the time-dependent noise schedule that uniquely determines the process, and $\tau$ denotes the rescaled time with respect to $\sigma_t$. A key observation is that the logarithm map $\exp^{-1}$ represents the direction of the shortest path between the current state and the endpoint while the magnitude of the drift increases to infinity with a rate $\sigma_t^2/(\tau_T - \tau_t)$ as $t \to T$, forcing the process to converge to the endpoint.

By leveraging the short-time asymptotic behavior of the Brownian motion and the Girsanov theorem, we theoretically derive in Appendix A.2 that the diffusion process described by Eq. (3) actually

---

[1] Here we assume that the endpoint is not in the cut locus of the current state for the inverse to be well-defined.
[2] Our Logarithm bridge can be considered a special case of Thompson (2018) which we describe in A.2.

converges to the endpoint $z$ regardless of the prior distribution $\Gamma$. Notably, when $\mathcal{M}$ is a Euclidean space $\mathbb{R}^d$, the logarithm bridge process reduces to the well-known Euclidean Brownian bridge process. But for general manifolds, our Logarithm bridge process differs from the Brownian bridge process of Eq. (2) due to the difference in the drifts.

Although the Logarithm bridge process provides a simple solution for constructing the generative process on simple manifolds, for instance, sphere and torus, it is not the case for general manifolds as the logarithm map is not given in closed form and is costly to compute on the fly. However, we can bypass the difficulty by taking a different perspective for defining the direction toward the endpoint on the manifold. Specifically, inspired by Chen & Lipman (2023), we consider the path on the manifold that minimizes the spectral distance $d_w(\cdot, \cdot)$, which is defined as follows:

$$d_w(x,y)^2 = \sum_{i=1}^{\infty} w(\lambda_i)\big(\phi_i(x) - \phi_i(y)\big)^2, \tag{4}$$

where $\lambda_i$ and $\phi_i$ are the eigenvalues and eigenfunctions of the Laplace-Beltrami operator, respectively, and $w$ is a monotonically decreasing function. From the fact that $\nabla d_w(\cdot, z)^2$ describes the tangent vector with the direction that minimizes the spectral distance between the current state and the endpoint $z$, we introduce a new family of bridge processes, namely the *Spectral Bridge Process*:

$$\mathbb{Q}_{spec}^z \; : \; \mathrm{d}\boldsymbol{X}_t = -\frac{1}{2}\frac{\sigma_t^2}{\tau_T - \tau_t}\frac{\nabla_{\boldsymbol{X}_t}d_w(\boldsymbol{X}_t, z)^2}{\|\nabla_{\boldsymbol{X}_t}d_w(\boldsymbol{X}_t, z)\|_{\mathcal{M}}^2}\mathrm{d}t + \sigma_t\mathbf{B}_t^{\mathcal{M}}, \quad \boldsymbol{X}_0 \sim \Gamma. \tag{5}$$

Practically, we use the truncated spectral distance where the eigenvalues $\lambda_i$ and the eigenfunctions $\phi_i$ are computed only once in advance of training.

## 3.2 DIFFUSION MIXTURE ON RIEMANNIAN MANIFOLD

**Riemannian Diffusion Mixture** Having the bridge processes at hand, we now build a generative process on manifolds that transports a prior distribution to the data distribution. Extending the diffusion mixture framework (Peluchetti, 2021; Liu et al., 2023) to the Riemannian setting, we construct a generative process by mixing a collection of bridge processes on the manifold, $\{\mathbb{Q}^z : z \sim \Pi\}$ where $\Pi$ denotes the data distribution. To be specific, we derive a diffusion process that admits a marginal density $p_t$ which is equal to the mixture of marginal densities $p_t^z$ of the bridge processes, that is modeled by the following SDE (we provide a detailed derivation in Appendix A.3):

$$\mathrm{d}\boldsymbol{X}_t = \frac{\sigma_t^2}{\tau_T - \tau_t}\left[\int \eta^z(\boldsymbol{X}_t, t)\frac{p_t^z(\boldsymbol{X}_t)}{p_t(\boldsymbol{X}_t)}\,\Pi(\mathrm{dvol}_z)\right]\mathrm{d}t + \sigma_t\mathrm{d}\mathbf{B}_t^{\mathcal{M}}, \quad \boldsymbol{X}_0 \sim \Gamma, \tag{6}$$

where $\eta^z$ denotes the unscaled drift of the bridge process with endpoint $z$ that represents the direction of the tangent vector, for example, $\exp_{\boldsymbol{X}_t}^{-1}(z)$ for the Logarithm bridge.

From the geometrical viewpoint, the drift of the mixture process corresponds to the weighted mean of tangent vectors heading in the direction of the endpoints in the data distribution as illustrated in Figure 1. Thus simulating the mixture process can be interpreted as taking a small step in the tangent space toward the direction of the most likely endpoint of the process. From this perspective, we can obtain an explicit prediction for the mixture process of the Logarithm bridges by projecting the drift onto the manifold along the geodesic using the exponential map as follows:

$$\hat{\boldsymbol{X}}_t := \exp_{\boldsymbol{X}_t}\left(\int \exp_{\boldsymbol{X}_t}^{-1}(z)\frac{p_t^z(\boldsymbol{X}_t)}{p_t(\boldsymbol{X}_t)}\,\Pi(\mathrm{dvol}_z)\right), \tag{7}$$

which corresponds to the most probable endpoint of the mixture process given the current state. This differentiates our method from the previous diffusion models that do not admit straightforward predictions for their denoising process on non-Euclidean manifolds.

Notably, by the construction of the mixture process, its terminal distribution is guaranteed to be equal to the data distribution $\Pi$ regardless of the initial distribution $\Gamma$. Thus our framework can be trivially applied to the case of an arbitrary prior distribution, which is not true for previous diffusion models (Bortoli et al., 2022; Huang et al., 2022) as they require careful design of the potential $U(\cdot)$ in the noising process (Eq. (1)). The difference comes from the fact that our mixture process does not rely on the time-reversal approach used for deriving denoising diffusion processes. This further yields freedom for the choice of $\sigma_t$ as it does not need to be decreasing or starting from a large noise scale.

When the mixture consists of the Logarithm bridges or the Spectral bridges, we refer to the mixture processes as the *Logarithm Bridge Mixture* (LogBM) and *Spectral Bridge Mixture* (SpecBM), respectively. Note that our LogBM generalizes the Euclidean diffusion mixture (Peluchetti, 2021; Liu et al., 2023) since the Logarithm bridge recovers the Brownian bridge for the Euclidean space.

**Probability Flow ODE** For a mixture process $\mathbb{Q}_f$, there exists a deterministic process that admits the same marginal densities, i.e., the probability flow (Maoutsa et al., 2020; Song et al., 2021). We can derive the probability flow by considering the time-reversed process $\mathbb{Q}_b$ of $\mathbb{Q}_f$ in two different perspectives: First, since $\mathbb{Q}_f$ is the mixture of bridges from $\Gamma$ to $\Pi$, $\mathbb{Q}_b$ corresponds to a mixture process built from the collection of time-reversed bridge processes from $\Pi$ to $\Gamma$, which can be represented by a SDE similar to Eq. (6). On the other hand, as the time-reversed process $\mathbb{Q}_b$ can be derived in terms of the score function (Theorem 3.1 of Bortoli et al. (2022)), we can represent the score function from the drifts of $\mathbb{Q}_f$ and $\mathbb{Q}_b$. As a result, the probability flow associated with the mixture process $\mathbb{Q}_f$ is obtained as the following ODE (see Appendix A.4 for detailed derivation):

$$\frac{\mathrm{d}}{\mathrm{d}t}\boldsymbol{Y}_t = \frac{1}{2}\Big(\eta_f(\boldsymbol{Y}_t, t) - \eta_b(\boldsymbol{Y}_t, T{-}t)\Big), \quad \boldsymbol{Y}_0 \sim \Gamma, \tag{8}$$

where $\eta_f$ and $\eta_b$ denote the drift of $\mathbb{Q}_f$ and $\mathbb{Q}_b$, respectively, and the likelihood of the probability flow can be computed as follows:

$$\log p_T(\boldsymbol{Y}_T) = \log p_0(\boldsymbol{Y}_0) + \frac{1}{2}\int_0^T \mathrm{div}\Big(\eta_f(\boldsymbol{Y}_t, t) - \eta_b(\boldsymbol{Y}_t, T{-}t)\Big)\mathrm{d}t, \tag{9}$$

We further discuss in Appendix A.4 that the probability flow derived from our mixture process is different from the continuous flows used in previous CNF models.

## 3.3 Two-way Bridge Matching

In order to use the mixture process as a generative model, we parameterize the drifts of the mixture process and its time-reversed process with neural networks, i.e., $\boldsymbol{s}_f^\theta(z, t) \approx \eta_f(z, t)$, $\boldsymbol{s}_b^\phi(z, t) \approx \eta_b(z, t)$. However, the drifts of the mixture processes cannot be directly approximated since we do not have access to the integral of Eq. (6). In what follows, we derive a simple and efficient training objective that is applicable to general manifolds without computing the Riemannian divergence.

The KL divergence between a mixture process $\mathbb{Q} : \mathrm{d}\boldsymbol{Z}_t = \eta(\boldsymbol{Z}_t, t)\mathrm{d}t + \nu_t\mathrm{d}B_t^{\mathcal{M}}$ with terminal distribution $\mathbb{Q}_T$ and its parameterized process $\mathbb{P}^\psi : \mathrm{d}\boldsymbol{Z}_t = s^\psi(\boldsymbol{Z}_t, t)\mathrm{d}t + \nu_t\mathrm{d}B_t^{\mathcal{M}}$ can be obtained from the Girsanov theorem as follows (we provide detailed derivation in Appendix A.5):

$$D_{KL}(\mathbb{Q}_T\|\mathbb{P}_T^\psi) \le D_{KL}(\mathbb{Q}\|\mathbb{P}^\psi) = \mathbb{E}_{\substack{z\sim\mathbb{Q}_T, \\ \boldsymbol{Z}\sim\mathbb{Q}^z}}\left[\frac{1}{2}\int_0^T \Big\|\nu_t^{-1}\Big(\boldsymbol{s}^\psi(\boldsymbol{Z}_t, t) - \eta^z(\boldsymbol{Z}_t, t)\Big)\Big\|_{\mathcal{M}}^2 \mathrm{d}t\right] + C. \tag{10}$$

where $\mathbb{P}_T^\psi$ denotes the terminal distributions of $\mathbb{P}^\psi$, $\mathbb{Q}^z$ and $\eta^z$ denotes the bridge process conditioned to $z$ and its drift, and $C$ is a constant. Unlike the Euclidean case, the trajectories $\boldsymbol{Z}_t$ of the bridge $\mathbb{Q}^z$ should be obtained through the simulation of the bridge process since the transition density of the Brownian motion is not accessible for general manifolds. Although we can use Eq. (10) to estimate the drifts, it is computationally expensive as the simulation of the bridge processes requires a large number of discretized steps due to the increasing magnitude of their drift near the terminal time.

We address this issue by proposing an efficient yet accurate training scheme, which we refer to as the *two-way bridge matching*. The main idea is to exploit the fact that the simulation of the bridge process can be performed from both forward and backward directions. Moreover, as the drifts $\eta_f$ and $\eta_b$ can be approximated independently, the trajectory can be obtained from a single bridge process with a fixed starting and endpoint instead of simulating two different bridge processes, reducing the computational cost in half. Altogether, two-way bridge matching can be formalized as follows:

$$\mathbb{E}_{\substack{(x,y)\sim(\Pi,\Gamma), \\ \boldsymbol{Z}\sim\mathbb{Q}^{x,y}}}\frac{1}{2}\int_0^T \sigma_t^{-2}\left[\Big\|\boldsymbol{s}_f^\theta(\boldsymbol{Z}_t, t) - \eta_f^x(\boldsymbol{Z}_t, t)\Big\|_{\mathcal{M}}^2 + \Big\|\boldsymbol{s}_b^\phi(\boldsymbol{Z}_t, T{-}t) - \eta_b^y(\boldsymbol{Z}_t, T{-}t)\Big\|_{\mathcal{M}}^2\right]\mathrm{d}t, \tag{11}$$

where $\mathbb{Q}^{x,y}$ denotes the bridge process with fixed starting point $x$ and endpoint $y$, and the trajectory $\boldsymbol{Z}_t$ is obtained by simulating the forward direction of $\mathbb{Q}^{x,y}$ if $t < t^*$ and otherwise from the backward

direction. Notably, from the result of Eq. (10), minimizing Eq. (11) guarantees to minimize the KL divergence between data distribution and terminal distribution of our parameterized mixture process.

However, we empirically observe that using Eq. (11) introduces high variance during training. Therefore, we present an equivalent objective that enables stable training by leveraging importance sampling. We use a proposal distribution $q(t) \propto \sigma_t^{-2}$ to adjust the weighting in Eq. (11) from $\sigma_t^{-2}$ to a constant, resulting in the following objective that is equivalent to Eq. (11):

$$\mathbb{E}_{\substack{(x,y)\sim(\Pi,\Gamma),\\t\sim q}}\mathbb{E}_{\boldsymbol{Z}_t\sim\mathbb{Q}^{x,y}}\left[\left\|\boldsymbol{s}_f^\theta(\boldsymbol{Z}_t,t)-\eta_f^x(\boldsymbol{Z}_t,t)\right\|_{\mathcal{M}}^2+\left\|\boldsymbol{s}_b^\phi(\boldsymbol{Z}_t,T-t)-\eta_b^y(\boldsymbol{Z}_t,T-t)\right\|_{\mathcal{M}}^2\right], \quad (12)$$

which we refer to as *time-scaled* two-way bridge matching. This can be interpreted as putting more weight on learning the drift from certain time intervals instead of considering them all equally. During training with Eq. (12), we first sample $t \sim q$, $x \sim \Pi$, and $y \sim \Gamma$ and then simulate $\boldsymbol{Z}_t \sim \mathbb{Q}^{x,y}$ using the two-way approach, where different $(t, x, y)$ are sampled to compute the expectation.

In particular, due to the time-scaled distribution $q$ in Eq. (12), our time-scaled two-way bridge matching differs from the Flow Matching objective which regresses the conditional vector field over uniformly distributed time. We experimentally validate the importance of the time-scaled distribution in Section 5.5, where using a uniform time distribution results in a significant drop in performance compared to using our time-scaled distribution. This is because Eq. (12) is guaranteed to minimize the KL divergence between data and terminal distribution of our parameterized process, whereas it is not true for a simple regression over uniformly distributed time. While the idea of the importance sampling for the time distribution was also used in Huang et al. (2022), our approach leverages a simple and easy-to-sample proposal distribution $q$, which is effective in stabilizing the training and improving the generation quality, without the need for additional computation or training time.

We empirically validate that our two-way approach can obtain accurate trajectories with significantly reduced simulation steps compared to the one-way simulation, resulting in up to $\times 34.9$ speed up for training. Furthermore, we show in Section 5 that the in-training simulation is not a significant overhead during training since the two-way approach greatly reduces the number of simulation steps without sacrificing the accuracy, which is significantly faster than the implicit score matching of previous models and comparable to Flow Matching on simple geometries.

**Connection with Riemannian Flow Matching**    Especially, in the case when the noise schedule is set to be very small, i.e., $\sigma_t \to 0$, we can recover the deterministic flow of Riemannian Flow Matching (RFM) (Chen & Lipman, 2023) from our mixture process, since the bridge processes with $\sigma_t \to 0$ correspond to the conditional vector fields of Flow Matching. Thereby RFM can be considered a special case of our framework when the randomness is removed from the mixture process.

However, stochasticity is crucial for learning the density on manifolds with non-trivial curvature. While obtaining a trajectory of the probability path for RFM during training requires a large number of simulation steps, we can obtain trajectories from the mixture process with only a few simulation steps thanks to its stochastic nature, which we empirically show in Section 5.5. Thus the existence of stochasticity dramatically reduces the number of in-training simulation steps compared to RFM, achieving $\times 12.8$ speed up in training without sacrificing the performance. We demonstrate in Figure 3 and 8 that our method is able to model complex distribution on the manifold with only a few in-training simulation steps, whereas RFM completely fails in such a setting. Furthermore, we can leverage the Girsanov theorem to derive that our training objective in Eq. (12) is guaranteed to minimize the KL divergence between the data distribution and the terminal distribution of our parameterized process, which does not apply to RFM as it is based on a deterministic process Additionally, the noise schedule provides flexible choices for designing the mixture process as we do not have any restrictions for $\sigma_t$, thus advantageous for modeling distributions on diverse manifolds.

## 4    RELATED WORK

**Euclidean Diffusion Models**    Diffusion models (Song & Ermon, 2019; Ho et al., 2020; Song et al., 2021) model the generative process via the denoising diffusion process derived from the time-reversal of the noising process. Recent works (Peluchetti, 2021; Liu et al., 2023; Peluchetti, 2023) introduce an alternative approach for modeling the generative process without using the time-reversal, namely diffusion mixture, by building bridge processes between the initial and the terminal distributions.

Table 1: **Test NLL results on earth and climate science datasets**. We report the mean of 5 different runs with different data splits. Best performance and its comparable results ($p > 0.05$) from the t-test are highlighted.

| Dataset size | Volcano 827 | Earthquake 6120 | Flood 4875 | Fire 12809 |
|---|---|---|---|---|
| RCNF (Mathieu & Nickel, 2020) | $-6.05 \pm 0.61$ | $0.14 \pm 0.23$ | $1.11 \pm 0.19$ | $-0.80 \pm 0.54$ |
| Moser Flow (Rozen et al., 2021) | $-4.21 \pm 0.17$ | $-0.16 \pm 0.06$ | $0.57 \pm 0.10$ | $-1.28 \pm 0.05$ |
| CNFM (Ben-Hamu et al., 2022) | $-2.38 \pm 0.17$ | $\mathbf{-0.38} \pm 0.01$ | $\mathbf{0.25} \pm 0.02$ | $-1.40 \pm 0.02$ |
| RFM (Chen & Lipman, 2023) | $-7.93 \pm 1.67$ | $-0.28 \pm 0.08$ | $0.42 \pm 0.05$ | $-1.86 \pm 0.11$ |
| StereoSGM (Bortoli et al., 2022) | $-3.80 \pm 0.27$ | $-0.19 \pm 0.05$ | $0.59 \pm 0.07$ | $-1.28 \pm 0.12$ |
| RSGM (Bortoli et al., 2022) | $-4.92 \pm 0.25$ | $-0.19 \pm 0.07$ | $0.45 \pm 0.17$ | $-1.33 \pm 0.06$ |
| RDM (Huang et al., 2022) | $-6.61 \pm 0.96$ | $\mathbf{-0.40} \pm 0.05$ | $0.43 \pm 0.07$ | $-1.38 \pm 0.05$ |
| Ours (LogBM) | $\mathbf{-9.52} \pm 0.87$ | $-0.30 \pm 0.06$ | $0.42 \pm 0.08$ | $\mathbf{-2.47} \pm 0.11$ |

| | Glycine (2D) 13283 | Proline (2D) 7634 | RNA (7D) 9478 |
|---|---|---|---|
| Dataset size | | | |
| MoPS | $2.08 \pm 0.009$ | $0.27 \pm 0.008$ | $4.08 \pm 0.368$ |
| RDM | $1.97 \pm 0.012$ | $\mathbf{0.12} \pm 0.011$ | $-3.70 \pm 0.592$ |
| RFM | $1.90 \pm 0.055$ | $0.15 \pm 0.027$ | $\mathbf{-5.20} \pm 0.067$ |
| Ours (LogBM) | $\mathbf{1.89} \pm 0.056$ | $\mathbf{0.14} \pm 0.027$ | $\mathbf{-5.27} \pm 0.090$ |

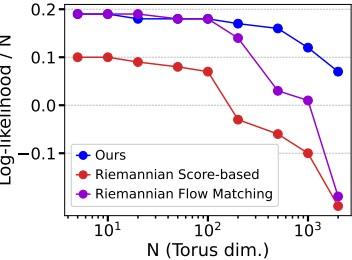

Figure 2: **(Left) Test NLL results on protein datasets**. Best performance and its comparable results ($p > 0.05$) from the t-test are highlighted in bold. **(Right) Comparison on high-dimensional tori**. We compare the log-likelihood in bits against RSGM and RFM where the results are obtained by running the open-source codes.

However, these methods are limited to Euclidean space and sub-optimal for modeling the data living on manifolds, for example, sphere for climate data and tori for biological data such as proteins. Our work extends the diffusion mixture framework to manifolds that generalizes the Euclidean case.

**Generative Models on Riemannian Manifolds**     Previous generative models (Gemici et al., 2016; Rezende et al., 2020; Bose et al., 2020) on Riemannian manifolds relied on projecting a Euclidean space to manifolds which is problematic since such mapping cannot be bijective, resulting in numerical instabilities. Recent works address this problem by constructing a mapping on the manifold that describes the transport from a prior distribution to the data distribution, namely the diffusion models and the continuous normalizing flow (CNF) models.

Bortoli et al. (2022) extends the score-based model to the manifold, while Huang et al. (2022) introduces a variational framework for diffusion models on manifold. However, score matching on the manifold either needs to be approximated or scales poorly to higher dimensions. Specifically, denoising score matching requires the conditional score function which is intractable even on simple manifolds and should rely on an approximation that obstructs exact training. Further, implicit score matching requires the computation of the Riemannian divergence which scales poorly to high-dimensional manifolds, while using the Hutchinson estimator (Hutchinson, 1989) introduces high variance in training. Moreover, previous diffusion models cannot be readily adapted to arbitrary prior distributions as defining a noising process that converges to a specific distribution other than the uniform or normal distribution requires a careful design that is non-trivial. In contrast, our framework provides efficient and scalable training that does not require divergence and does not rely on approximations of the heat kernel. Since the construction of the mixture process guarantees convergence to the data distribution regardless of the prior distribution, our method can be readily extended for arbitrary prior distribution. We further compare with Diffusion Schrödinger Bridge (Thornton et al., 2022) and recent works focusing on specific geometries in Appendix A.6.

On the other hand, CNF models build a continuous-time flow (Chen et al., 2018; Grathwohl et al., 2019) on the manifold by parameterizing the vector field. However, previous CNF models (Lou et al., 2020; Mathieu & Nickel, 2020; Falorsi & Forré, 2020) rely on simulation-based maximum likelihood training which is computationally expensive. Recent works (Rozen et al., 2021; Ben-Hamu et al., 2022) introduce simulation-free training methods on simple geometries, but they scale poorly to high-dimension due to the computation of the divergence and further cannot be adapted to non-simple geometries. Chen & Lipman (2023) extends the Flow Matching framework (Lipman et al., 2023) to manifolds which learns the probability path by regressing the conditional vector fields. Instead of the deterministic flow, our work constructs a generative process using diffusion processes for

Figure 3: **Visualization of the generated samples and the learned density** of our method and RFM on the mesh datasets. Blue dots denote the generated samples and darker colors of red indicate higher likelihood.

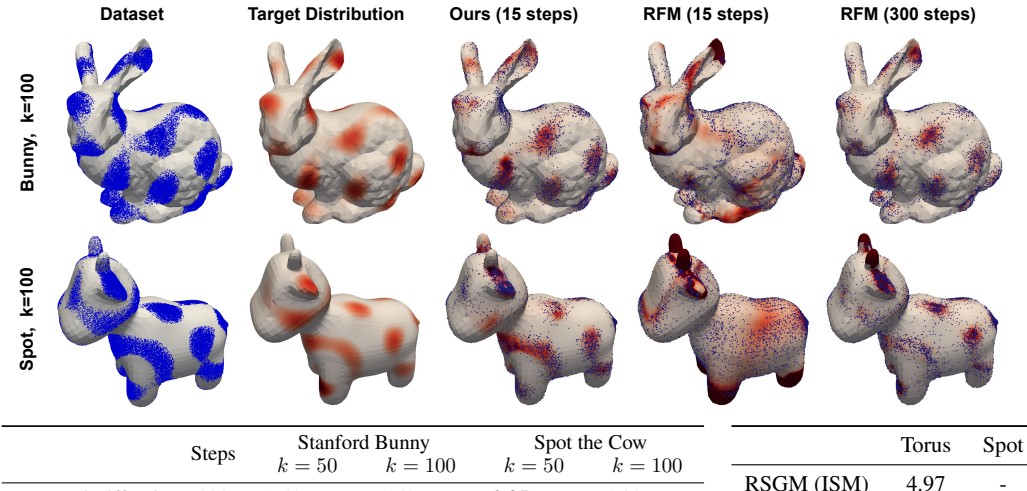

Table 2: **(Left) Test NLL results on mesh datasets**. We report the mean of 5 different runs. Best performance and its comparable results ($p > 0.05$) from the t-test are highlighted. **(Right) Comparison of the training time**. We report the relative training time of the baselines with respect to ours on high-dimensional torus and Spot.

| | Steps | Stanford Bunny | | Spot the Cow | |
| --- | --- | --- | --- | --- | --- |
| | | $k = 50$ | $k = 100$ | $k = 50$ | $k = 100$ |
| RFM w/ Diff. Dist. | 300 | 1.48 ± 0.01 | 1.53 ± 0.01 | **0.95** ± 0.05 | 1.08 ± 0.05 |
| RFM w/ Bihar. Dist. | 300 | 1.55 ± 0.01 | 1.49 ± 0.01 | **1.08** ± 0.05 | 1.29 ± 0.05 |
| Ours w/ Diff. Dist. | 15 | **1.42** ± 0.01 | **1.41** ± 0.00 | 0.99 ± 0.03 | **0.97** ± 0.03 |
| Ours w/ Bihar. Dist. | 15 | 1.55 ± 0.02 | 1.45 ± 0.01 | **1.09** ± 0.06 | **0.97** ± 0.02 |

| | Torus | Spot |
| --- | --- | --- |
| RSGM (ISM) | 4.97 | - |
| RSGM (SSM) | 1.20 | - |
| RFM | 0.97 | 12.83 |
| Ours | 1.00 | 1.00 |

which stochasticity is crucial for learning on general geometries, as it dramatically reduces in-training simulation steps while achieving superior performance compared to the CNF model.

## 5 EXPERIMENTS

We experimentally validate our method on diverse datasets including real-world benchmarks as well as synthetic distributions. We follow the experimental settings of previous works (Bortoli et al., 2022; Chen & Lipman, 2023) where we provide the details of the training setup in Appendix B. We compare our method against generative models on manifolds: **RCNF** (Mathieu & Nickel, 2020), **Moser Flow** (Rozen et al., 2021), **CNFM** (Ben-Hamu et al., 2022) and **RFM** (Chen & Lipman, 2023) are CNF models, **StereoSGM** (Bortoli et al., 2022) is a Euclidean score-based model using stereographic projection, **RSGM** (Bortoli et al., 2022) is a Riemannian score-based model, and **RDM** (Huang et al., 2022) is a Riemannian diffusion model based on a variational framework.

### 5.1 EARTH AND CLIMATE SCIENCE DATASETS

We evaluate the generative models on real-world datasets living on the 2-dimensional sphere, which consists of earth and climate science events including volcanic eruptions (NOAA, 2020b), earthquakes (NOAA, 2020a), floods (Brakenridge, 2017), and wild fires (EOSDIS, 2020). We use the LogBM, i.e. a mixture of logarithm bridges in Eq. (3), as the geodesic is accessible in a closed form on the sphere. Table 1 demonstrates that our method significantly outperforms the baselines on the Volcano and the Fire datasets that require high fidelity as the data are concentrated in specific regions. Ours constantly outperforms the Riemannian score-based model while matching the performance of RFM on the Earthquake and the Flood dataset. We visualize the generated samples and learned densities of our model in Figure 4 showing that our method is capable of capturing the distribution on the sphere. We further compare the convergence of the generative processes measured by the geodesic distance in Figure 5 where the generative process of our method converges faster than that of the baselines. We observe that the prediction from our model (Eq. (7)) converges faster than that of RFM, indicating that ours is able to make more accurate predictions during the generation process.

### 5.2 PROTEIN DATASETS

We further experiment on protein datasets represented on $n$-dimensional torus from their torsion angles, consisting of 500 high-resolution proteins (Lovell et al., 2003) and 113 selected RNA

sequences Murray et al. (2003) preprocessed by Huang et al. (2022) which we provide details in Appendix B.3. We additionally compare our method against the Mixture of Power Spherical (MoPS) (De Cao & Aziz, 2020) which models the distribution as a mixture of power spherical distributions. We use the LogBM as the geodesic is accessible in closed form on the torus. Table of Figure 2 demonstrates that ours outperforms or is on par with RDM while making marginal improvements over RFM, where the baselines are likely to be close to optimal. We provide the results of the other two protein datasets (General, Pre-Pro) in Table 3 showing comparable results with RFM. We visualize the learned density in Figure 7 where our model properly fits the data distribution.

## 5.3 HIGH-DIMENSIONAL TORI

We validate the scalability of our method using the synthetic data on high-dimensional tori. We follow Bortoli et al. (2022) by creating a wrapped Gaussian distribution with a random mean and variance of 0.2 on $n$-dimensional tori where we compare the performance with RSGM trained via the implicit score matching and RFM. To make a fair comparison, we use the same model architecture for all methods, where the total number of parameters for our models match that of the baselines. We describe the detailed setting in Appendix B.4. As shown in Figure 2 (Right), ours constantly outperforms RSGM, especially in high dimensions where RSGM scales poorly with the dimensions due to the high variance in computing the stochastic divergence of the training objective. RFM also shows a significant drop in high dimensions which implies that the vector field could not be well-approximated with a limited number of parameters. On the other hand, ours is able to scale fairly well even for high dimensions as our training objective of Eq. (12), which guarantees to minimize the KL divergence between the data and model distribution, does not require computation of divergence. In particular, as shown in Table 2 (Right), we achieve up to $\times 5$ speedup in training compared to RSGM using implicit score matching (ISM), and also significantly faster than RSGM using ISM with the stochastic estimator (SSM). Our training time is comparable to Flow Matching which is simulation-free on tori, as we use only 15 steps for the in-training simulation.

## 5.4 GENERAL CLOSED MANIFOLDS

To validate that our framework can be applied to general manifolds with non-trivial curvature, we evaluate on synthetic distributions on triangular meshes. Following Chen & Lipman (2023), we construct the target distribution on Stanford Bunny (Turk & Levoy, 1994) and Spot the Cow (Crane et al., 2013) from the $k$-th eigenfunction of the mesh, which we provide detail in Appendix B.5. We use SpecBM, i.e. a mixture of spectral bridges in Eq. (5), with either the diffusion distance (Coifman & Lafon, 2006) or the biharmonic distance (Lipman et al., 2010). As visualized in Figure 3 and Figure 8, our method is able to fit the complex distributions using only 15 steps for the in-training simulation, while RFM completely fails when using a small number of in-training simulation steps. We show in Table 2 that our method outperforms RFM by using only 5% of in-training simulation steps required for RFM while achieving $\times 12.8$ speed up in training compared to RFM.

## 5.5 FURTHER ANALYSIS

**Non-Compact Manifold**   To validate that our framework can be applied to non-compact manifolds, we experiment on the synthetic distributions on a 2-dimensional hyperboloid. Figure 9 demonstrates that our method is able to model the target distributions.

**Time-scaled Training Objective**   We empirically validate that the time-scaled objective of Eq. (12) is crucial for learning the distribution, by comparing ours with a variant trained with uniform time distribution similar to Flow Matching. Table 6 shows that the variant results in a significant drop in performance compared to our time-scaled objective.

**Number of In-Training Simulation Steps**   We empirically demonstrate that our method can be trained using only 15 steps for the in-training simulation: We show in Figure 10 (a) and (c) that the trajectories of the mixture process simulated with 15 steps result in an almost similar distribution to the trajectories simulated with 500 steps, which cannot be achieved with the one-way simulation as shown in Figure 10 (b). Especially, Figure 10 (d) demonstrates that using a small noise scale, resembling a deterministic process, requires a large number of simulation steps to obtain accurate trajectories on triangular mesh, explaining the reason for the failure of RFM in Figure 3 and 8.

## 6 CONCLUSION

In this work, we present Riemannian Diffusion Mixture, a new framework for the generative modeling on manifolds that builds a generative diffusion process on the manifold as a mixture of Riemannian bridge processes, instead of using the time-reversal approach of previous diffusion models. Our method enables efficient training on general manifolds that does not require computing the divergence that can scale to high dimensions. Our approach dramatically reduces the number of in-training simulation steps while showing superior performance on diverse manifold datasets. Our work provides a promising direction for the diffusion models on manifold which could be applied to various scientific fields, for example, the design of proteins.

## REPRODUCIBILITY STATEMENT

We use Pytorch (Paszke et al., 2019) and JAX (Bradbury et al., 2018) to implement our method, which we have included our codes in the supplementary material. We have specified experimental details in Section B.

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

# A  DERIVATIONS

## A.1  DIFFUSION PROCESS ON RIEMANNIAN MANIFOLD

Brownian bridge process is a diffusion process described by the Brownian motion conditioned to a fixed starting point and endpoint, which is induced by the following infinitesimal generator:

$$\frac{1}{2}\Delta_{\mathcal{M}} + \nabla \log p(\cdot, z, T - t), \tag{13}$$

where $p$ is the transition density of the Brownian motion, i.e. the heat kernel defined on $\mathcal{M}$, and $z$ denotes the fixed endpoint. Thus Brownian bridge process can be modeled by the following SDE:

$$\mathbb{Q}_{bb}^z : \mathrm{d}\boldsymbol{X}_t = \nabla_{\boldsymbol{X}_t} \log p_{\mathcal{M}}(\boldsymbol{X}_t, z, T-t)\mathrm{d}t + \mathrm{d}\mathbf{B}_t^{\mathcal{M}}. \tag{14}$$

We refer the readers to Hsu (2002) for a formal definition of the Brownian bridge process. The theoretical properties of Brownian bridge have been studied by previous works (Hsu, 1990; Driver, 1994).

One can construct a diffusion process that converges to a stationary distribution described by the Langevin dynamics with respect to the Brownian motion $\mathbf{B}_t^{\mathcal{M}}$:

$$\mathrm{d}\boldsymbol{X}_t = -\frac{1}{2}\nabla_{\boldsymbol{X}_t} U(\boldsymbol{X}_t)\mathrm{d}t + \mathrm{d}\mathbf{B}_t^{\mathcal{M}}, \tag{15}$$

where the terminal distribution satisfies $\mathrm{d}p(x)/\mathrm{dvol}_x \propto e^{-U(x)}$ (Durmus, 2016) for a potential function $U$. For example, $U(x) = d_g(x, \mu)^2/(2\gamma^2) + \log\|D\exp_\mu^{-1}(x)\|$ results in a stationary distribution equal to the wrapped Gaussian distribution with an arbitrary mean location $mu \in \mathcal{M}$, where $d_g$ denotes the geodesic distance.

## A.2  LOGARITHM BRIDGE PROCESS

Here we show that our proposed Logarithm bridge of Eq. (3) describes a diffusion process that converges to an endpoint. For the notational simplicity, we omit the subscript $\mathcal{M}$ for the heat kernel on $\mathcal{M}$. First, we derive that the following simplified process is a bridge process with an endpoint $z$:

$$\mathbb{Q}_{log}^z : \mathrm{d}\boldsymbol{X}_t = \frac{1}{T-t}\exp_{\boldsymbol{X}_t}^{-1}(x)\mathrm{d}t + \mathrm{d}\mathbf{B}_t^{\mathcal{M}}. \tag{16}$$

For any pair of points $x$ and $y$ on $\mathcal{M}$, the following short-time asymptotic of the heat kernel holds (Theorem 5.2.1 of (Hsu, 2002)):

$$\lim_{t\to 0} t \log p(x, y, t) = -\frac{d_g(x, y)^2}{2}, \tag{17}$$

where $d_g(\cdot, \cdot)$ is a geodesic distance. Further leveraging the identity from Proposition 6 of McCann (2001), we obtain the following result:

$$\lim_{t\to 0} t\nabla_x \log p(x, y, t) = -\frac{1}{2}\nabla_x d_g(x, y)^2 = \exp_x^{-1}(y). \tag{18}$$

Furthermore, we have an upper bound for the gradient of the logarithmic heat kernel for any pair of points $x, y \in \mathcal{M}$ and $t \in [0, T]$ as follws (Theorem 5.5.3 of (Hsu, 2002)):

$$\left\|\nabla_x \log p(x, y, t)\right\|_{\mathcal{M}} \le C\left[\frac{d_g(x, y)}{t} + \frac{1}{\sqrt{t}}\right] \tag{19}$$

where $C$ is a constant. From Eq. (19), we have the following bound:

$$\left\| \frac{\exp_x^{-1}(z)}{T-t} - \nabla_x \log p\left(x, z, T-t\right) \right\|_{\mathcal{M}} \tag{20}$$

$$= \mathbb{1}_{\{t > T-\epsilon\}} \cdot \left\| \frac{\exp_x^{-1}(z)}{T-t} - \nabla_x \log p\left(x, z, T-t\right) \right\|_{\mathcal{M}} \tag{21}$$

$$+ \mathbb{1}_{\{t \leq T-\epsilon\}} \cdot \left\| \frac{\exp_x^{-1}(z)}{T-t} - \nabla_x \log p\left(x, z, T-t\right) \right\|_{\mathcal{M}} \tag{22}$$

$$\leq \delta(\epsilon) + \frac{1}{\epsilon} \left\| \exp_x^{-1}(z) \right\|_{\mathcal{M}} + \left\| \nabla_x \log p(x, z, T-t) \right\|_{\mathcal{M}} \tag{23}$$

$$\leq \delta(\epsilon) + \frac{1}{\epsilon} d_g(x, y) + C \left[ \frac{d_g(x, y)}{\epsilon} + \frac{1}{\sqrt{\epsilon}} \right], \tag{24}$$

which holds for every $\epsilon > 0$ where $\delta(t)$ denotes the term of Eq. (21). Since $\delta(\epsilon) \to 0$ as $\epsilon \to 0$ from the results of Eq. (18), we obtain the following:

$$\left\| \frac{\exp_x^{-1}(z)}{T-t} - \nabla_x \log p\left(x, z, T-t\right) \right\|_{\mathcal{M}} < \infty. \tag{25}$$

Finally, using the Girsanov theorem for $\mathbb{Q}_{bb}^z$ and $\mathbb{Q}_{log}^z$, we obtain the following result:

$$D_{KL}\left(\mathbb{Q}_{bb}^z \| \mathbb{Q}_{log}^z\right) = \frac{1}{2} \mathbb{E}_{\boldsymbol{X} \sim \mathbb{Q}_{bb}^z} \left[ \int_0^T \left\| \frac{\exp_{\boldsymbol{X}_t}^{-1}(z)}{T-t} - \nabla_x \log p\left(\boldsymbol{X}_t, z, T-t\right) \right\|_{\mathcal{M}}^2 \mathrm{d}t \right] < \infty, \tag{26}$$

which implies that the Brownian bridge and the Logarithm bridge have the same support. Thereby, the Logarithm bridge is actually a bridge process that converges to a fixed endpoint $z$.

The Logarithm bridge process presented in Eq. (3) is the result of using the change of time (Øksendal, 2003) on Eq. (16) with respect to the rescaled time $\tau(t) \coloneqq \int_0^t \sigma_s^2 \mathrm{d}s$. Furthermore, similar to the Euclidean bridge processes introduced in Wu et al. (2022), the derivation of our Logarithm bridge process provides a more general form of bridge processes on the manifold:

$$\mathrm{d}\boldsymbol{X}_t = \left[ \frac{\sigma_t^2}{\tau_T - \tau_t} \exp_{\boldsymbol{X}_t}^{-1}(z) + \sigma_t \nabla_{\boldsymbol{X}_t} U(\boldsymbol{X}_t, t) \right] \mathrm{d}t + \sigma_t \mathrm{d}\mathbf{B}_t^{\mathcal{M}}, \tag{27}$$

for a scalar function $U$ that satisfies $\mathbb{E}_{\boldsymbol{X} \sim \mathbb{Q}^{z,bb}} \int_0^T \|\nabla U(\boldsymbol{X}_t, t)\|_{\mathcal{M}}^2 < \infty$ which is sufficient for bounded functions $U$.

Note that Thompson (2018) introduces a general family of bridge processes, namely the Fermi bridge, that describes diffusion processes conditioned to a submanifold $N$, and has been used for the simulation of bridge processes (Jensen & Sommer, 2021). One could derive our Logarithm bridge from the Fermi bridge by constraining $N$ to a single point, but to the best of our knowledge, our Logarithm bridge was not studied or provided in previous works, and further leveraging the Logarithm bridge process in the context of generative modeling is new.

## A.3 DIFFUSION MIXTURE ON RIEMANNIAN MANIFOLD

In this section, we extend the diffusion mixture representation (Peluchetti, 2021) to the Riemannian setting. We start with the statement of the diffusion mixture representation. For a collection of diffusion processes $\{\mathbb{Q}^\lambda : \lambda \in \Lambda\}$ on $\mathcal{M}$ modeled by the following SDEs:

$$\mathbb{Q}^\lambda : \mathrm{d}\boldsymbol{X}_t^\lambda = \eta^\lambda(\boldsymbol{X}_t^\lambda, t)\mathrm{d}t + \sigma_t^\lambda \mathrm{d}\mathbf{B}_t^\lambda, \ \boldsymbol{X}_0^\lambda \sim p_0^\lambda \tag{28}$$

where $\mathbf{B}_t^\lambda$ are independent Brownian motions on $\mathcal{M}$ and $p_0^\lambda$ denotes the initial distributions of $\mathbb{Q}^\lambda$. Denoting the marginal distribution of $\mathbb{Q}^\lambda$ as $p_t^\lambda$ and a mixing distribution $\mathcal{L}$ on $\Lambda$, consider the following density and the distribution defined as the mixture of $p_t^\lambda$ and $p_0^\lambda$, respectively, as follows:

$$p_t(x) = \int p_t^\lambda(x)\mathcal{L}(\mathrm{dvol}_x), \ p_0(x) = \int p_0^\lambda(x)\mathcal{L}(\mathrm{dvol}_x). \tag{29}$$

Then there exists a diffusion process on $\mathcal{M}$ such that its marginal density is equal to $p_t$ with the initial distribution given as $p_0$, described by the following SDE:

$$\mathbb{Q}^* : \mathrm{d}\boldsymbol{X}_t^\lambda = \eta(\boldsymbol{X}_t^\lambda, t)\mathrm{d}t + \sigma_t \mathrm{d}\mathbf{B}_t^\lambda, \ \boldsymbol{X}_0^\lambda \sim p_0, \tag{30}$$

where the drift $\eta_t$ and the diffusion coefficient $\sigma_t$ satisfy the following:

$$\eta(x,t) = \int \eta^\lambda(x,t) \frac{p_t^\lambda(x)}{p_t(x)} \mathcal{L}(\mathrm{dvol}_x), \ \ \sigma_t^2 = \int (\sigma_t^\lambda)^2 \frac{p_t^\lambda(x)}{p_t(x)} \mathcal{L}(\mathrm{dvol}_x). \tag{31}$$

The proof for the Riemannian setting extends that of the Euclidean case, where we leverage the Fokker-Planck equation to characterize the marginal density. From the condition of Eq. (31), we can derive the following:

$$\frac{\partial p_t(x)}{\partial t} = \int \frac{\partial}{\partial t} p_t^\lambda(x) \mathcal{L}(\mathrm{dvol}_x) \tag{32}$$

$$= \int \left[ -\mathrm{div}\left(p_t^\lambda(x)\eta^\lambda(x,t)\right) + \frac{1}{2}(\sigma_t^\lambda)^2 \Delta_{\mathcal{M}} p_t^\lambda(x) \right] \mathcal{L}(\mathrm{dvol}_x) \tag{33}$$

$$= -\mathrm{div}\left( p_t(x) \int \eta^\lambda(x,t)\frac{p_t^\lambda(x)}{p_t(x)} \mathcal{L}(\mathrm{dvol}_x) \right) + \frac{1}{2}\Delta_{\mathcal{M}}\left( p_t(x) \int (\sigma_t^\lambda)^2 \frac{p_t^\lambda(x)}{p_t(x)} \mathcal{L}(\mathrm{dvol}_x) \right) \tag{34}$$

$$= -\mathrm{div}\left(p_t(x)\eta(x,t)\right) + \frac{1}{2}\sigma_t^2 \Delta_{\mathcal{M}} p_t(x) \tag{35}$$

where the second equality is derived from the Fokker-Planck equation with respect to the process $\mathbb{Q}^\lambda$. Since Eq. (35) corresponds to the Fokker-Planck equation with respect to the mixture process, we can conclude that $p_t$ is the marginal density of the mixture process.

## A.4 PROBABILITY FLOW ODE

Here we provide the derivation of the probability flow of the mixture process $\mathbb{Q}_f$ by considering its time-reversed process $\mathbb{Q}_b$ in two different perspectives. First, the time-reversed process $\mathbb{Q}_b$ corresponds to a mixture process built from the collection of time-reversed bridge processes from $\Pi$ to $\Gamma$, which can be modeled by the following SDE:

$$\mathrm{d}\overline{\boldsymbol{X}}_t = \frac{\sigma_{T-t}^2}{\tau_T - \tau_{T-t}} \left[ \int \eta_b^z(\overline{\boldsymbol{X}}_t, t) \frac{p_t^z(\overline{\boldsymbol{X}}_t)}{p_t(\overline{\boldsymbol{X}}_t)} \Gamma(\mathrm{dvol}_z) \right] \mathrm{d}t + \sigma_{T-t}\mathrm{d}\mathbf{B}_t^{\mathcal{M}}, \ \ \overline{\boldsymbol{X}}_0 \sim \Pi, \tag{36}$$

where $\tau$ is defined in Eq. (3). On the other hand, $\mathbb{Q}_b$ can be derived in terms of the score function from Theorem 3.1 of Bortoli et al. (2022) as follows:

$$\mathrm{d}\overline{\boldsymbol{X}}_t = \left[ -\eta_f(\overline{\boldsymbol{X}}_t, t) + \sigma_{T-t}^2 \nabla_{\overline{\boldsymbol{X}}_t} \log p_t(\overline{\boldsymbol{X}}_t) \right]\mathrm{d}t + \sigma_{T-t}\mathrm{d}\mathbf{B}_t^{\mathcal{M}}. \tag{37}$$

Therefore, we can obtain the score function in terms of the drifts of $\mathbb{Q}_f$ and $\mathbb{Q}_b$ as follows:

$$\nabla \log p_t(\boldsymbol{X}_t) = \left(\eta_f(\boldsymbol{X}_t, t) + \eta_b(\boldsymbol{X}_t, T-t)\right)/\sigma_t^2, \tag{38}$$

and as a result, the probability flow associated with the mixture process $\mathbb{Q}_f$ is obtained by the following ODE:

$$\frac{\mathrm{d}}{\mathrm{d}t}\boldsymbol{Y}_t = \left(\eta_f(\boldsymbol{Y}_t, t) - \frac{1}{2}\nabla \log p_t(\boldsymbol{Y}_t)\right) = \frac{1}{2}\left(\eta_f(\boldsymbol{Y}_t, t) - \eta_b(\boldsymbol{Y}_t, T-t)\right), \ \ \boldsymbol{Y}_0 \sim \Gamma, \tag{39}$$

Using Proposition 2 of Mathieu & Nickel (2020), we can compute the likelihood of the probability flow as follows:

$$\log p_T(\boldsymbol{Y}_T) = \log p_0(\boldsymbol{Y}_0) + \frac{1}{2}\int_0^T \mathrm{div}\left(\eta_f(\boldsymbol{Y}_t, t) - \eta_b(\boldsymbol{Y}_t, T-t)\right)\mathrm{d}t, \tag{40}$$

It is worth noting that the probability flow of the mixture process is different from the continuous flows used in previous works. This is due to the difference in the marginal densities that are characterized by different laws: By construction, the marginal density of the probability flow is equal to the marginal density of the associated mixture process which is described by the Fokker-Planck equation:

$$\frac{\partial p_t(z)}{\partial t} = -\mathrm{div}\left(\eta_f(z,t)p_t(z)\right) + \frac{1}{2}\sigma_t^2 \Delta_{\mathcal{M}} p_t(z), \tag{41}$$

whereas the marginal density of a deterministic process is described by the transportation equation:

$$\frac{\partial \tilde{p}_t(z)}{\partial t} = -\mathrm{div}\left(\eta_{\mathrm{CNF}}(z,t)\tilde{p}_t\right). \tag{42}$$

## A.5 BRIDGE MATCHING ON RIEMANNIAN MANIFOLD

We first derive the KL divergence between a mixture process $\mathbb{Q} : \mathrm{d}\boldsymbol{Z}_t = \eta(\boldsymbol{Z}_t, t)\mathrm{d}t + \nu_t \mathrm{d}B_t^{\mathcal{M}}$ with terminal distribution $\mathbb{Q}_T$ and its parameterized process $\mathbb{P}^\psi : \mathrm{d}\boldsymbol{Z}_t = s^\psi(\boldsymbol{Z}_t, t)\mathrm{d}t + \nu_t \mathrm{d}B_t^{\mathcal{M}}$ by leveraging the Girsanov theorem as follows:

$$D_{KL}(\mathbb{Q}_T \| \mathbb{P}_T^\psi) \leq D_{KL}(\mathbb{Q} \| \mathbb{P}^\psi) = \mathbb{E}_{\substack{z \sim \mathbb{Q}_T, \\ \boldsymbol{Z} \sim \mathbb{Q}^z}} \left[ \log \frac{\mathrm{d}\mathbb{Q}^z}{\mathrm{d}\mathbb{P}^\psi}(\boldsymbol{Z}) + \log \frac{\mathrm{d}\mathbb{Q}}{\mathrm{d}\mathbb{Q}^z}(\boldsymbol{Z}) \right] \tag{43}$$

$$= \mathbb{E}_{z \sim \mathbb{Q}_T} \left[ D_{KL}(\mathbb{Q}^z \| \mathbb{P}^\psi) \right] + C_1 \tag{44}$$

$$= \mathbb{E}_{\substack{z \sim \mathbb{Q}_T, \\ \boldsymbol{Z} \sim \mathbb{Q}^z}} \left[ \frac{1}{2} \int_0^T \left\| \nu_t^{-1} \Big( \boldsymbol{s}^\psi(\boldsymbol{Z}_t, t) - \eta^z(\boldsymbol{Z}_t, t) \Big) \right\|_{\mathcal{M}}^2 \mathrm{d}t \right] + C_2 \tag{45}$$

where the first inequality is from the data processing inequality. Note that this result is a straightforward extension of the bridge matching (Liu et al., 2022) to the Riemannian setting. Leveraging the fact that the time-reversed process of the mixture process $\mathbb{Q}_f$ is also a mixture process of time-reversed bridge processes (Eq. (36)), the models $\boldsymbol{s}_f^\theta$ and $\boldsymbol{s}_b^\phi$ can be trained to approximate the drifts $\eta_f$ and $\eta_b$, respectively, with the following objectives:

$$\mathcal{L}_f(\theta) = \mathbb{E}_{\substack{x \sim \Pi, \\ \boldsymbol{X} \sim \mathbb{Q}_f^x}} \left[ \frac{1}{2} \int_0^T \left\| \sigma_t^{-1} \Big( \boldsymbol{s}_f^\theta(\boldsymbol{X}_t, t) - \eta_f^x(\boldsymbol{X}_t, t) \Big) \right\|_{\mathcal{M}}^2 \mathrm{d}t \right], \tag{46}$$

$$\mathcal{L}_b(\phi) = \mathbb{E}_{\substack{y \sim \Gamma, \\ \overline{\boldsymbol{X}} \sim \mathbb{Q}_b^y}} \left[ \frac{1}{2} \int_0^T \left\| \sigma_{T-t}^{-1} \Big( \boldsymbol{s}_b^\phi(\overline{\boldsymbol{X}}_t, t) - \eta_b^y(\overline{\boldsymbol{X}}_t, t) \Big) \right\|_{\mathcal{M}}^2 \mathrm{d}t \right]. \tag{47}$$

Instead of separately training the models by simulating two different bridge processes $\mathbb{Q}_f^x$ and $\mathbb{Q}_b^y$, we can train the models simultaneously by simulating a single bridge process $\mathbb{Q}^{x,y}$ with a fixed starting point $x$ and endpoint $y$, reducing the computational cost for the simulation in half. Furthermore, when simulating $\mathbb{Q}^{x,y}$, we introduce a two-way approach to obtaining the trajectory $\boldsymbol{Z}_t$ from $\mathbb{Q}^{x,y}$ simulated from $x$ to $y$ for $t < t^*$ and otherwise simulated from $y$ to $x$, which results in the following loss presented in Eq. (11):

$$\mathbb{E}_{\substack{(x,y) \sim (\Pi, \Gamma), \\ \boldsymbol{Z} \sim \mathbb{Q}^{x,y}}} \frac{1}{2} \int_0^T \sigma_t^{-2} \left[ \left\| \boldsymbol{s}_f^\theta(\boldsymbol{Z}_t, t) - \eta_f^x(\boldsymbol{Z}_t, t) \right\|_{\mathcal{M}}^2 + \left\| \boldsymbol{s}_b^\phi(\boldsymbol{Z}_t, T-t) - \eta_b^y(\boldsymbol{Z}_t, T-t) \right\|_{\mathcal{M}}^2 \right] \mathrm{d}t. \tag{48}$$

Leveraging an importance sampling with a proposal distribution $q(t) \propto \sigma_t^{-2}$, we obtain the time-scaled two-way bridge matching:

$$\mathbb{E}_{\substack{(x,y) \sim (\Pi, \Gamma), \\ \boldsymbol{Z} \sim \mathbb{Q}^{x,y}}} \mathbb{E}_{t \sim q} \left[ \left\| \boldsymbol{s}_f^\theta(\boldsymbol{Z}_t, t) - \eta_f^x(\boldsymbol{Z}_t, t) \right\|_{\mathcal{M}}^2 + \left\| \boldsymbol{s}_b^\phi(\boldsymbol{Z}_t, T-t) - \eta_b^y(\boldsymbol{Z}_t, T-t) \right\|_{\mathcal{M}}^2 \right]. \tag{49}$$

## A.6 COMPARISON WITH DIFFUSION MODELS ON MANIFOLD

Here we discuss further the comparison with diffusion models on manifolds, extending Section 4. Thornton et al. (2022) extends Diffusion Schrödinger Bridge to the manifold setting, which aims to find the forward and backward processes between distributions that minimize the KL divergence to the Brownian motion. However, Thornton et al. (2022) uses Iterative Proportional Fitting to alternatively train the models that require computing the divergence for numerous iterations, which is computationally expensive compared to our divergence-free two-way bridge matching that is able to train the models simultaneously. Finding the solution to Schrödinger bridge problem as in Thornton et al. (2022) brings additional complexity in the context of generative models compared to our method of directly learning the mixture process. Recent works focus on specific geometries such as SO(3) (Leach et al., 2022), SE(3) (Yim et al., 2023; Urain et al., 2023), and product of tori (Jing et al., 2022), or constrained manifolds (Fishman et al., 2023) defined by set of inequality constraints.

## A.7 SAMPLING

With the trained models $\boldsymbol{s}_f^\theta$ and $\boldsymbol{s}_b^\phi$ that estimate the drifts of the mixture process and its reversed process respectively, we can sample from the parameterized mixture process using only $\boldsymbol{s}_f^\theta$ as follows:

$$\mathrm{d}\boldsymbol{X}_t = \boldsymbol{s}_f^\theta(\boldsymbol{X}_t, t)\mathrm{d}t + \sigma_t \mathrm{d}\mathbf{B}_t^{\mathcal{M}}, \tag{50}$$

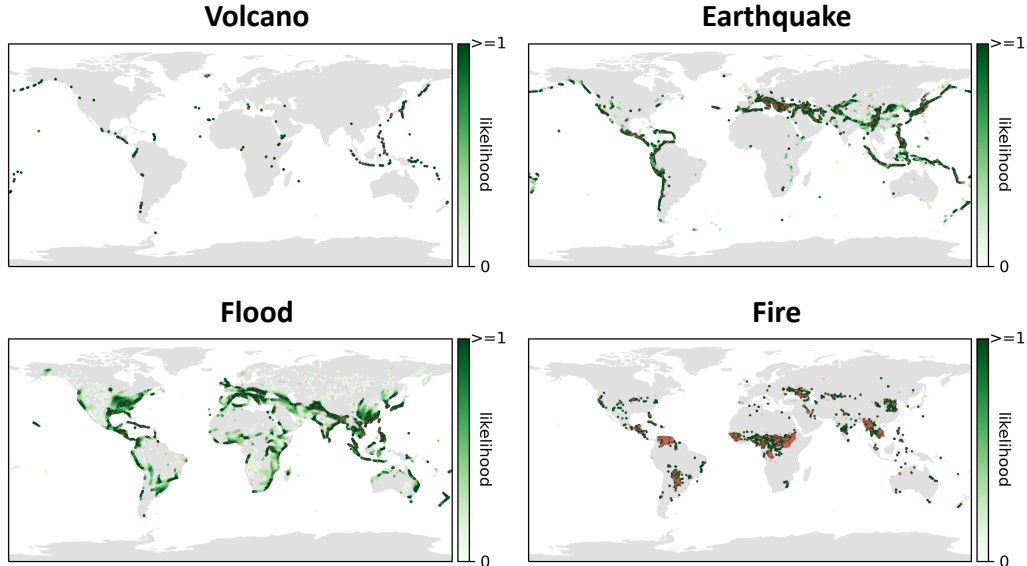

Figure 4: **Visualization of the generated samples and learned density** of our model on earth and climate science datasets. Red dots denote samples from the test set and green dots denote the generated samples, where darker colors indicate higher likelihood.

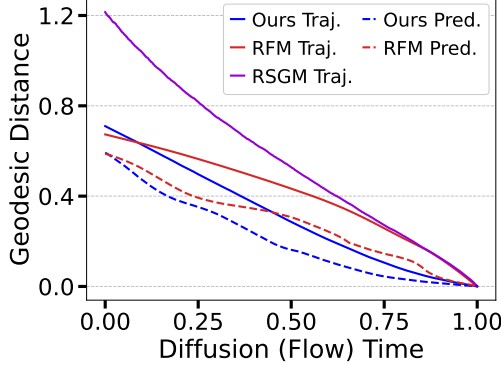

Figure 5: **Convergence of the generative processes** on the Volcano dataset. We compare the geodesic distance between the trajectory (Traj.) and the final sample of each method. We further compare the convergence of the predictions (Pred.) made by ours and that of RFM.

| Ours | Volcano | Flood |
|---|---|---|
| Uniform | -5.37 $\pm$ 0.67 | 0.67 $\pm$ 0.14 |
| Time-scaled | **-9.52** $\pm$ 0.87 | **0.42** $\pm$ 0.08 |

Figure 6: **Ablation study on the time-scaled training objective**. We report the test NLL of our method (Time-scaled) against a variant trained with uniformly distributed time (Uniform), instead of the time-scaled distribution $q$ in Eq. (12).

which can be simulated by leveraging the Geodesic Random Walk (Jørgensen, 1975; Bortoli et al., 2022). Further, we can solve the parameterized probability flow ODE of Eq. (8) using both models $\boldsymbol{s}_f^\theta$ and $\boldsymbol{s}_b^\phi$ as follows:

$$\frac{\mathrm{d}}{\mathrm{d}t}\boldsymbol{X}_t = \frac{1}{2}\left(\boldsymbol{s}_f^\theta(\boldsymbol{X}_t, t) - \boldsymbol{s}_b^\phi(\boldsymbol{X}_t, T-t)\right), \tag{51}$$

with existing ODE solvers. Note that the marginal density of the probability flow ODE is equal to the marginal density of the corresponding mixture process by construction. Furthermore, since the construction of the diffusion mixture guarantees that the initial distribution and the terminal distribution are equal to the prior distribution and the data distribution respectively, the computed likelihood of the parameterized probability ODE coincides with the likelihood of the parameterized mixture process assuming the models closely approximates the drifts of the mixture process and its reversed process.

Table 3: **Test NLL results on protein datasets**. We report the mean of 5 different runs with different data splits. Best performance and its comparable results ($p > 0.05$) from the t-test are highlighted in bold.

| Dataset size | General (2D) 138208 | Glycine (2D) 13283 | Proline (2D) 7634 | Pre-Pro (2D) 6910 | RNA (7D) 9478 |
|---|---|---|---|---|---|
| Mixture of Power Spherical | $1.15 \pm 0.002$ | $2.08 \pm 0.009$ | $0.27 \pm 0.008$ | $1.34 \pm 0.019$ | $4.08 \pm 0.368$ |
| Riemannian Diffusion Model | $1.04 \pm 0.012$ | $1.97 \pm 0.012$ | $\mathbf{0.12} \pm 0.011$ | $1.24 \pm 0.004$ | $-3.70 \pm 0.592$ |
| Riemannian Flow Matching | $\mathbf{1.01} \pm 0.025$ | $\mathbf{1.90} \pm 0.055$ | $0.15 \pm 0.027$ | $\mathbf{1.18} \pm 0.055$ | $\mathbf{-5.20} \pm 0.067$ |
| Ours (LogBM) | $\mathbf{1.01} \pm 0.026$ | $\mathbf{1.89} \pm 0.056$ | $\mathbf{0.14} \pm 0.027$ | $\mathbf{1.18} \pm 0.059$ | $\mathbf{-5.27} \pm 0.090$ |

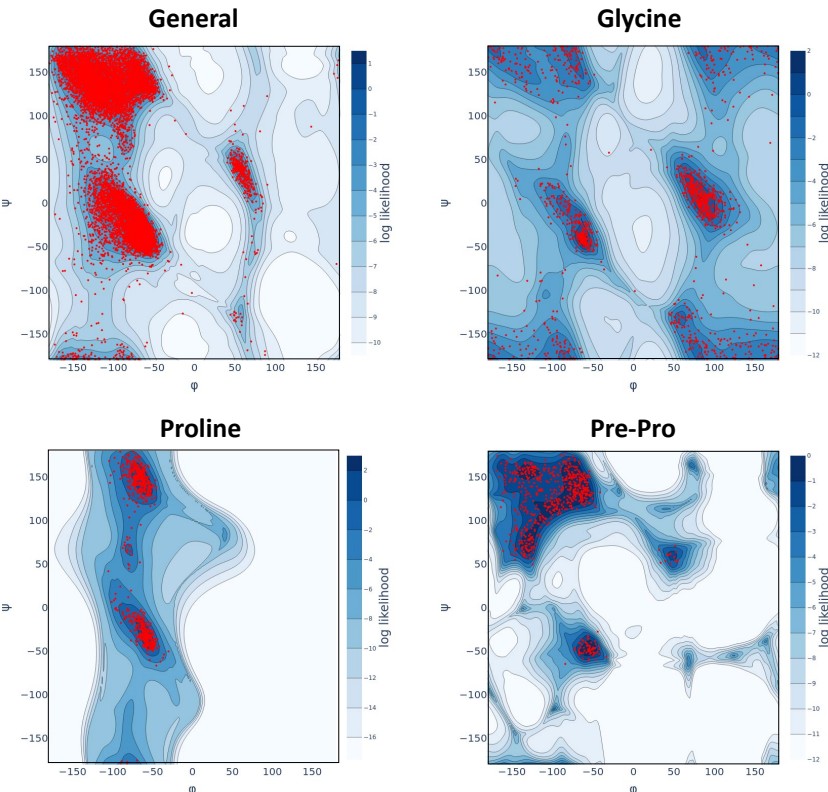

Figure 7: **Visualization of the learned density** of our model on protein datasets using the Ramachandran contour plots. The red dots denote the samples from the test set and the blue color denotes the log-likelihood where the darker color indicates higher likelihood.

# B    EXPERIMENTAL DETAILS

## B.1    IMPLEMENTATION DETAILS

We follow the experimental settings of previous works (Bortoli et al., 2022; Chen & Lipman, 2023) including the data splits with the same seed values of 0-4 for five different runs. We split the datasets into training, validation, and test sets with (0.8, 0.1, 0.1) proportions. Following Chen & Lipman (2023), we use the validation NLL for early stopping and the test NLL is computed from the checkpoint that achieved the best validation NLL. We parameterize the drifts of the mixture processes with multilayer perceptrons where we concatenate the time to the input, following the previous works. For all experiments except the high dimensional tori, we use 512 hidden units and select the number of layers from 6 to 13, and use either the sinusoidal or swish activation function. All models are trained with Adam optimizer and we either do not use a learning rate scheduler or use the scheduler where the learning rate is annealed with a linear map and then applying cosine scheduler, as introduced in Bortoli et al. (2022). We use the exponential moving average for the model weights Polyak & Juditsky (1992) with decay 0.999.

Figure 8: **Visualization of the generated samples and the learned density** of our method and RFM on the mesh datasets. Blue dots denote the generated samples and darker colors indicate higher likelihood.

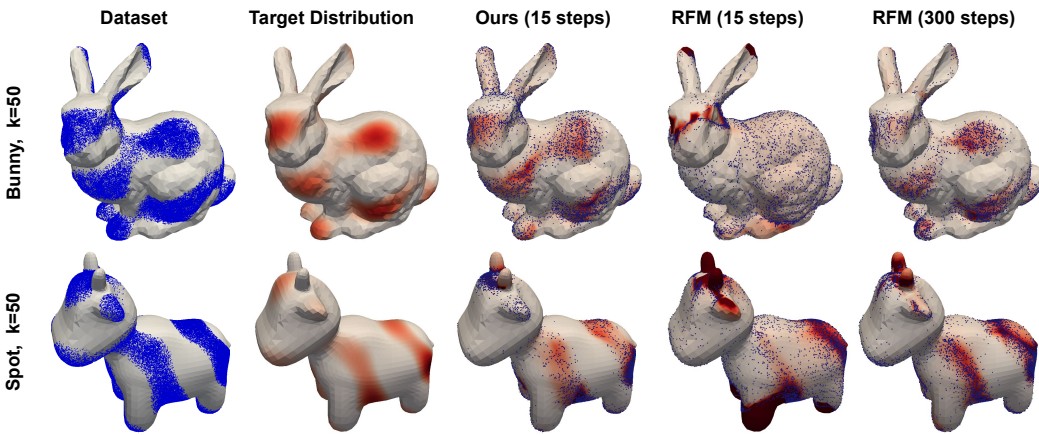

We parameterize the drifts of the mixture processes in the ambient space with projection onto the tangent space as follows:

$$\boldsymbol{s}^\theta(x,t) = \mathrm{proj}_x(\tilde{\boldsymbol{s}}^\theta(x,t)). \tag{52}$$

where $\mathrm{proj}_x$ is a orthogonal projection onto the tangent space at $x$. For all experiments, we train our models using the time-scaled two-way bridge matching in Eq. (12), where we use 15 steps for the in-training simulation carried out by Geodesic Random Walk (Jørgensen, 1975; Bortoli et al., 2022).

Except for the mesh experiments, we compute the likelihood of our parameterized probability flow ODE using Dormand-Prince solver (Dormand & Prince, 1980) with absolute and relative tolerance of $1e-5$, following the previous works (Bortoli et al., 2022; Chen & Lipman, 2023). For the mesh experiments, we compute the likelihood with 1000 Euler steps with projection after every step as done in Chen & Lipman (2023).

For all experiments, we use NVIDIA GeForce RTX 3090 and 2080 Ti and implement the source code with PyTorch (Paszke et al., 2019) and JAX (Bradbury et al., 2018).

### B.2   EARTH AND CLIMATE SCIENCE DATASETS

We follow the data splits of previous works (Bortoli et al., 2022; Chen & Lipman, 2023), reporting an average of five different runs with different data splits using the same seed values of 0-4. For a fair comparison with baselines, we set the prior distribution to be a uniform distribution on the sphere.

The convergence analysis demonstrated in Figure 5 was conducted on the models trained on the Volcano dataset, where we measure the geodesic distance between the final sample and the trajectory $\boldsymbol{Z}_t$ of each method for discretized time steps $t$. The convergence of the predictions was also measured similarly, where we use the parameterized prediction of Eq. (7).

### B.3   PROTEIN DATASETS

We follow the experimental setup of Huang et al. (2022) and Chen & Lipman (2023) where we use the dataset compiled by Huang et al. (2022) that consists of 500 high-resolution proteins (Lovell et al., 2003) and 113 selected RNA sequences Murray et al. (2003). The proteins and the rnas are divided into monomers where the joint structures are removed and use the backbone conformation of the monomer. For proteins, this results in 3 torsion angles of the amino acid where the $180°$ angle is removed, thus the proteins data can be represented on 2D torus. For RNA, the 7 torsion angles are represented on 7D torus.

We follow the data splits of Chen & Lipman (2023) and report an average of five different runs with different data splits using the same seed values of 0-4. For a fair comparison with the baselines, we also set the prior distribution to be a uniform distribution on the 2D and 7D tori.

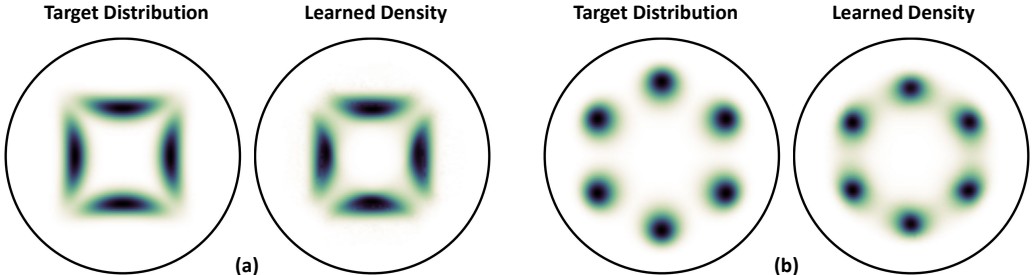

Figure 9: **Visualization of the learned density** of our model on the hyperboloid. We visualize the synthetic distributions and the learned density on the hyperboloid via projection onto a Poincare disk. (a) visualizes a mixture of four wrapped Gaussian distributions and (b) visualizes a mixture of six wrapped Gaussian distributions.

## B.4 HIGH-DIMENSIONAL TORI

We follow the experimental setup of Bortoli et al. (2022) where we create the dataset as a wrapped Gaussian distribution on a high dimensional tori with uniformly sampled mean and scale of 0.2. Since we evaluate on higher dimensions, up to 2000 dimensions, we use 2048 hidden units for all methods. Specifically, we use MLP with 3 hidden layers and 2048 hidden units for RSGM (Bortoli et al., 2022) and RFM (Chen & Lipman, 2023). To make a fair comparison with the baselines, we match the number of model parameters by using MLP with 2 hidden layers and 2048 hidden units for the model estimating the mixture process, i.e. $s_f^\theta$, and use MLP with 1 hidden layer and 512 hidden units for the model estimating the time-reversed mixture process, i.e. $s_b^\phi$. We train all methods for 50k iterations with a batch size of 512 without early stopping and evaluate the log-likelihood per dimension for 20k generated samples. We also set the prior distribution to be a uniform distribution on the high-dimensional tori. We measure the training time of each method implemented by JAX (Bradbury et al., 2018) for a fair comparison.

## B.5 GENERAL CLOSED MANIFOLDS

We use the triangular meshes provided by Chen & Lipman (2023): An open-source mesh is used for Spot the Cow and a downsampled mesh with 5000 triangles is used for Stanford Bunny, where the 3D coordinates of the meshes are normalized so that the points lie in $(0, 1)$. Following Chen & Lipman (2023), the target distributions on the mesh are created by first computing the $k$-th eigenfunction (associated with non-zero eigenvalue) on three times upsampled mesh, thresholding at zero, and then normalizing the resulting function. The visualization of generated samples and learned density of RFM in Figure 3 and 8 are obtained by running the open source code. For a fair comparison with RFM, we set the prior distribution to be a uniform distribution on the mesh. We measure the training time of ours and RFM implemented in JAX (Bradbury et al., 2018) for a fair comparison.

## B.6 FURTHER ANALYSIS

**Non-Compact Manifold**   We create the synthetic distributions on a 2-dimensional hyperboloid using a mixture of wrapped Gaussian distributions. We use MLP with 4 layers with 512 hidden units and trained for 100k iterations without early stopping. We visualize the learned density in Figure 9 by projecting onto a Poincare disk.

**Time-scaled Training Objective**   We compare our framework trained with Eq. (12) against a variant trained with uniformly distributed time instead of time-scaled distribution $q$ in Eq. (12), on the Volcano dataset. We follow the experimental setup of earth and climate science experiments.

**Number of In-Training Simulation Steps**   We measure the maximum mean discrepancy (MMD) (Gretton et al., 2012) between the simulated trajectory $X_t$ of the LogBM with 500 steps in a two-way approach, i.e. almost exact trajectory of the mixture process, and the simulated trajectories with a smaller number of steps. For Figure 10 (d), we use a very small noise scale for the mixture process to mimic a deterministic process. Note that the absolute scales of the MMD among Figure 10

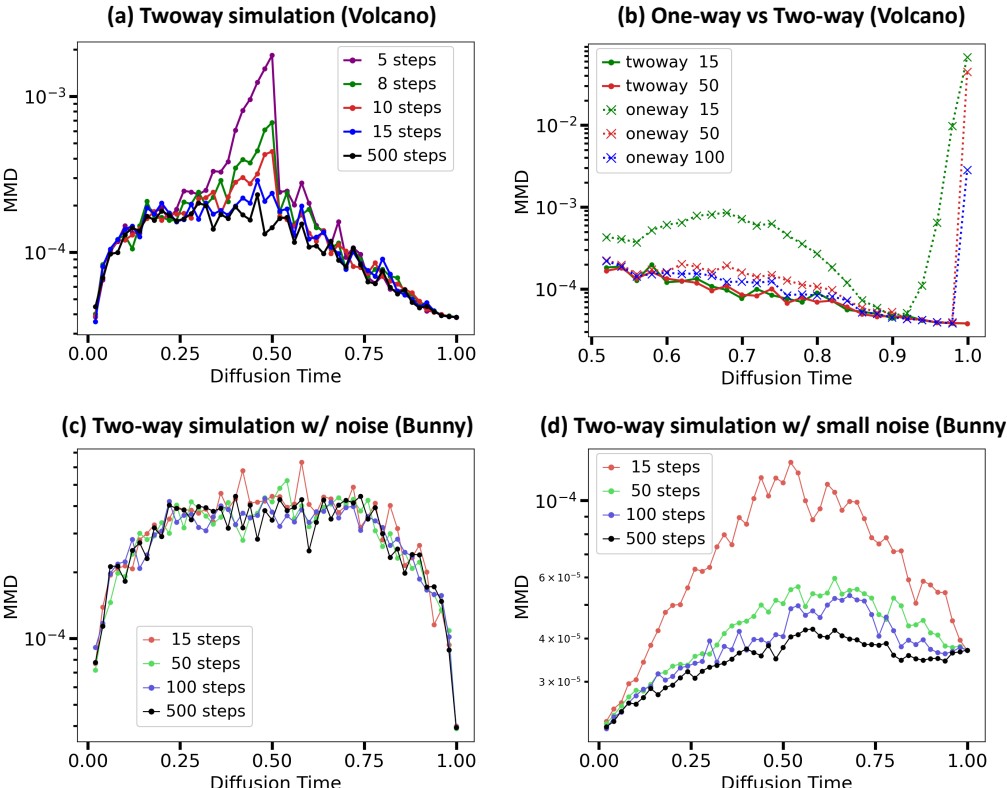

Figure 10: MMD distance between the simulated trajectories of of our Riemannian Bridge Mixture with 500 steps in a two-way approach, i.e. almost exact trajectories, and the simulated trajectories with a smaller number of steps. **(a)** We compare the MMD results on the Volcano dataset by differing the number of steps for the two-way approach where we observe that the MMD results of 15 steps are almost the same as the exact simulation. **(b)** We compare the MMD results with the one-way approach where we can see that the one-way approach requires a large number of steps to obtain accurate trajectories. **(c)** We compare the MMD results of the two-way approach on the Stanford Bunny dataset where we observe that 15 steps are enough to obtain accurate trajectories. **(d)** We compare the MMD results of the two-way approach when using a small noise scale for the mixture process, where using 15 steps produces highly inaccurate trajectories and requires more than 100 steps to obtain accurate trajectories.

(a)-(d) are not directly comparable, as the MMD are measured for different reference distributions. The MMD results should be interpreted as how much they deviate from the MMD result of the 'almost exact' trajectory (simulated by 500 steps).

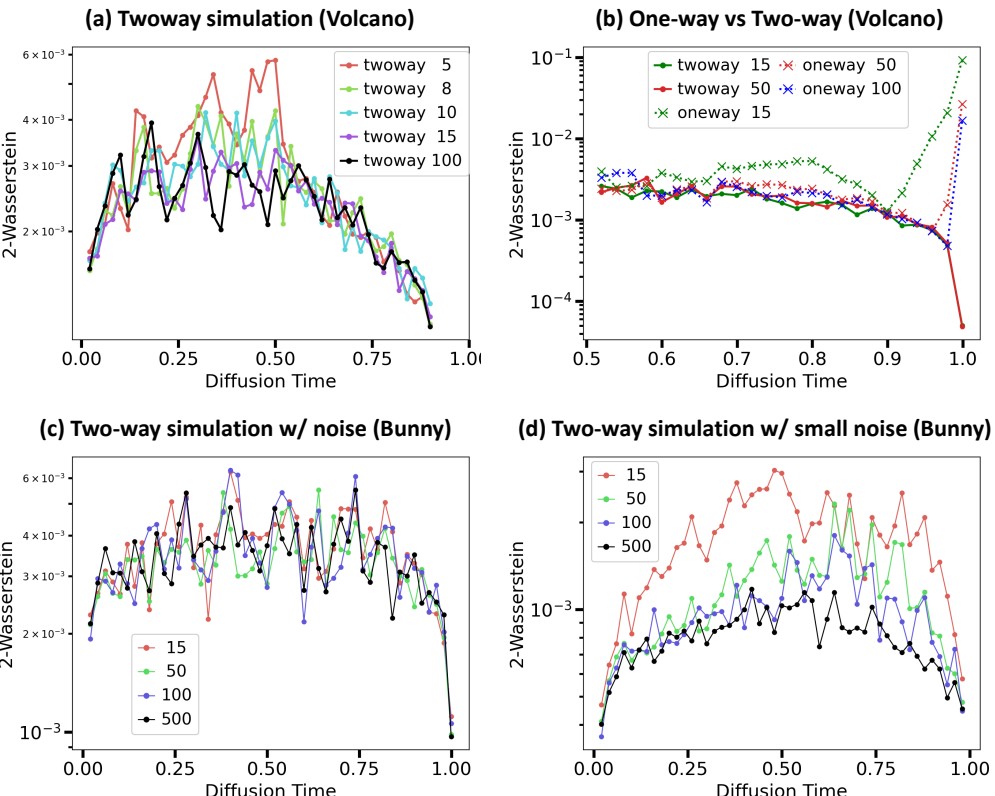

Figure 11: Wasserstein distance between the simulated trajectories of our Riemannian Bridge Mixture with 500 steps in a two-way approach, i.e. almost exact trajectories, and the simulated trajectories with a smaller number of steps. **(a)** We compare the distance results on the Volcano dataset by differing the number of steps for the two-way approach where we observe that the results of 15 steps are almost the same as the exact simulation. **(b)** We compare the distance results with the one-way approach where we can see that the one-way approach requires a large number of steps to obtain accurate trajectories. **(c)** We compare the distance results of the two-way approach on the Stanford Bunny dataset where we observe that 15 steps are enough to obtain accurate trajectories. **(d)** We compare the distance results of the two-way approach when using a small noise scale for the mixture process, where using 15 steps produces highly inaccurate trajectories and requires more than 100 steps to obtain accurate trajectories.

