# OpenReview forum: "Generative Modeling on Manifolds Through Mixture of Riemannian Diffusion Processes"
_ICLR.cc/2024/Conference — Submitted to ICLR 2024_

### Official Review · Reviewer_8EUf · 2023-10-24

**Soundness:** 3 good
**Presentation:** 2 fair
**Contribution:** 2 fair
**Rating:** 5
**Confidence:** 4

**Summary:**

The paper proposes to approach generative modelling on Riemannian manifolds by constructing a mixture of diffusion processes. This is achieved by defining a family of stochastic processes with explicit drift terms that mimic the Euclidean Brownian bridge drift, and mixing the drift according to the data distribution. The authors show how training can be achieved in this scheme.

**Strengths:**

- the authors approach diffusion generative modelling on manifolds with new ideas
- guiding a diffusion process by mixing tangent directions to the data distribution is a nice idea

**Weaknesses:**

- the presentation is a places not clear. For example, I have a hard time extracting meaning from a sentences such as "In addition, previous diffusion models on the manifold are geometrically not intuitive as their generative processes are derived from the time reversal of the noising process that accompanies score function in the drift which does not explicitly take into account the underlying geometry." I believe the score of a Brownian motion is indeed quite related to the underlying geometry?
- as far as I can read, the class of processes in equation (3) are not new (they are claimed to be new in the paper). In the stochastic analysis literature, they are denoted 'Fermi bridges' see https://arxiv.org/abs/1604.05182 They have been used for bridge simulation in e.g. https://arxiv.org/abs/2105.13190 and https://arxiv.org/abs/2104.03193 . I haven't seen them used in context of generative modelling though.
- I am unsure what is the main benefit of the approach: The score matching in denoising diffusion models is replaced by a loss depending on the distance which is also in general expensive to compute (the logarithm map (inverse exponential) on general manifolds is expensive to compute). I am not sure of the general validity of claims such as 'Our method shows superior performance on diverse manifolds that can scale to higher dimensions' because logarithm map computations can be quite non-trivial in high dimensions
- I don't think the authors convincingly argue why (11) is fast to evaluate. As I read it, it will require a large amount of logarithm evaluations during training. The spectral bridge processes can speed this up, but finding eigenfunctions of the Laplace-Beltrami operator on high-dimensional manifolds is likely not feasible

**Questions:**

I would be great to hear convincing counterarguments to the weaknesses listed above. I think the approach has merit, but the weakness listed above makes me unsure about the actual benefit of the method.

---

> ### Author Response · Authors · 2023-11-16
> **Initial Response to Reviewer 8EUf (1/2)**
>
> We sincerely thank you for your constructive and helpful comments. We appreciate your positive comments that
> - We approach diffusion generative modeling on manifolds with new ideas.
> - Guiding a diffusion process by mixing tangent directions to the data distribution is a nice idea.
>
> We provide an updated revision of the paper reflecting your comments (highlighted in orange) and we initially address all your concerns below.
>
> ---
>
> **Comment 1:**
> The meaning of the sentence “... score function in the drift which does not explicitly take into account the underlying geometry." is not clear.
>
> **Response 1:**
> The purpose of the statement is to explain that the **score function does not provide an explicit geometrical interpretation** for the diffusion of previous methods. In contrast, for our approach, the drift of the Logarithm bridge yields an intuitive geometrical explanation that corresponds to a tangent vector minimizing the geodesic distance toward an endpoint.
>
> We have clarified the statement as “... noising process that accompanies score function in the drift which does not provide explicit geometric interpretation.” in the updated revision.
>
> ---
>
> **Comment 2:**
> Class of processes in Eq. (3) is not new as they are denoted as Fermi bridges [1] and have been used for bridge simulation [2, 3].
>
> [1] Thompson, Brownian bridges to submanifolds, Potential Analysis 2018
>
> [2] Jensen and Sommer, Simulation of Conditioned Semimartingales on Riemannian Manifolds, arXiv 2021
>
> [3] Bui et al., Inference for partially observed Riemannian Ornstein-Uhlenbeck diffusions of covariance matrices, Bernoulli 2023
>
> **Response 2:**
> Thank you for the related works on diffusion bridges. However, Fermi bridge [1] describes a bridge process to a submanifold $N$ instead of a single point, and [1] or the related works do not study or present the explicit form of our Logarithm bridge. By constraining the submanifold $N$ to a single point, one can derive our Logarithm bridge from the Fermi bridge, but the derivation is not trivial and was not explored before, to the best of our knowledge.
>
> Furthermore, in our work, we provide the geometric interpretation for the drift of the Logarithm bridge, i.e., the tangent vector minimizing the geodesic distance toward its endpoint, which is not studied in [1]. Especially, based on this we propose a novel generative framework (Logarithm bridge mixture) which is a novel contribution in the context of generative modeling.
>
> We have added the comparison with the Fermi bridge in the updated revision (Appendix A.2). Note that we could not find a process similar to our Logarithm bridge or Fermi bridge in [3].

---

> ### Author Response · Authors · 2023-11-16
> **Initial Response to Reviewer 8EUf (2/2)**
>
> **Comment 3-1:**
> I am unsure what is the main benefit of the approach.
>
> **Response 3-1:**
> Many real-world problems we are interested in consider manifolds that satisfy either (1) have known geodesics, for example, robotics motion planning (SO(3)), protein modeling (SE(3)), neuroscience (SPD matrices), astrophysics (hypersphere),  and computational chemistry (product of tori), or (2) the eigenfunctions of the Laplace-Beltrami operator can be computed, for instance, manifolds represented as meshes. **When dealing with these real-world problems, our work provides practical benefits compared to previous works** as follows:
>
> - Our method achieves superior performance on diverse manifolds.
> - Our approach can scale to higher dimensions compared to previous methods that require computation of Riemannian divergence.
> - Our method has a significantly faster training time compared to previous works since ours is divergence-free and requires greatly reduced in-training simulation steps.
> - Our framework is readily adaptable to general manifolds and arbitrary prior distributions, which is not the case for previous methods.
> - Additionally, our work provides an intuitive geometrical interpretation of the diffusion process and further yields predictions of the final result (Eq. (7), Figure 5).
>
>
> ---
>
> **Comment 3-2:**
> Logarithm map on general manifolds can be expensive to compute and can be quite non-trivial in high dimensions.
>
> **Response 3-2:**
> Even though computing logarithm maps can be non-trivial and expensive for general high-dimensional manifolds, **many real-world problems (e.g., robotics, biology, chemistry) consider manifolds with known geodesics, and the logarithm maps are easy to compute even for high dimensions**. When dealing with these real-world problems, our framework has significant advantages in faster training and reduced computational cost, compared to previous works that require Riemannian divergence. Also, our method can scale to higher dimensions compared to previous methods as shown from the experiments on high-dimensional tori. Additionally, when the computation of the logarithm map is expensive and non-trivial, we can leverage the spectral bridge process (Eq. (5)) that bypasses computing the logarithm map as the reviewer thankfully acknowledges.
>
> We agree that when scaling to high-dimensional general manifolds for which computing the logarithm map is expensive, our framework can also have difficulty like previous works. We have toned down our claim as “our method can scale to higher dimensions when the geodesics are known.“ in the updated revision.
>
> ---
>
>
> **Comment 4:**
> I don't think the authors convincingly argue why Eq. (11) is fast to evaluate as it will require a large amount of logarithm evaluations during training.
>
> **Response 4:**
> In many real-world problems (e.g., robotics, biology, chemistry) we are interested in consider manifolds where the logarithm maps are known and easy to compute. When dealing with these real-world problems, our training objective of Eq. (12) can be computed significantly faster compared to the objectives of previous works (RCNF, RSGM, RDM), since **computing the logarithm map given in closed-form costs significantly less compared to computing the Riemannian divergence which requires evaluating the Jacobian (exact divergence) or the gradient (estimated divergence)**. We empirically show that our method achieves $\\times$5 speedup compared to RSGM.
>
> ---
> ---
>
> We thank the reviewer again for their time and feedback, and we hope that our responses have addressed any remaining questions.

---

> > ### Comment · Reviewer_8EUf · 2023-11-22
> > **Response**
> >
> > Thank you for the response. The response has not changed my rating (if anything for the worse). As the authors write, Fermi bridges are equivalent for single point N, and this has been used in other papers. Not mentioning Fermi bridges in the main part of the paper means relevant background work is not sufficiently covered.

---

> ### Author Response · Authors · 2023-11-21
> **Gentle Reminder for Reviewer 8EUf**
>
> Dear Reviewer 8EUf,
>
>
> Thank you for reviewing our paper. As the interactive discussion phase will end this Wednesday (22nd AOE), we politely ask you to check our responses. We summarize our responses as follows:
>
>
> - We modified the unclear sentence regarding the score function as “the score function does not provide an explicit geometrical interpretation”.
> - We made clear that the explicit form of our Logarithm bridge and its geometric interpretations as tangent directions were not studied in the previous works related to the Fermi bridge.
> - We highlighted the main benefits of our framework for solving real-world problems in diverse scientific fields.
> - We toned down our claim as “our method can scale to higher dimensions on manifolds with the known geodesics”.
> - We clarified that for many real-world problems considering manifolds with known geodesics, the logarithm maps can be efficiently computed even for high dimensions, and our training objective (Eq. (11)) can be evaluated significantly faster compared to computing the Riemannian divergence.
>
>
> We hope that our responses have addressed your concerns, and we hope you kindly consider updating the rating accordingly. Please let us know if there are any other things that we need to clarify or provide. We sincerely appreciate your valuable suggestions.
>
> Best regards,
> Authors

---

> ### Author Response · Authors · 2023-11-22
> **Thank you for the valuable comment**
>
> We sincerely appreciate your time and effort in reviewing our paper. We apologize for missing the related work, and we have referenced [1] in Section 3.1 when defining our Logarithm bridge process in the revised paper: we explained that our Logarithm bridge can be considered a special case of the Fermi bridge when restricting the submanifold $N$ to a single point. However, we could not find previous works that study the Fermi bridge with $N$ restricted to a single point, as previous works have focused on the case where $N$ is set to be the fiber in the Lie group [2] or the diagonal of the product manifold [3].
>
> We would like to emphasize that the **main contribution of our work is proposing a generative framework on general manifolds by leveraging the mixture of bridge processes** (Logarithm bridges and Spectral bridges) for which the drift is designed as the weighted mean of tangent directions to the data distribution. To the best of our knowledge, the Fermi bridges have not been used in the context of generative modeling in the previous works.
>
> We hope you kindly consider the novelty of our approach in the context of generative modeling and further the practical advantages of our method for evaluating our work. Thank you once again for reviewing our paper and providing valuable comments.
>
> Best regards,
> Authors
>
> ---
>
> [1] Thompson, Brownian bridges to submanifolds, Potential Analysis 2018
>
> [2] Jensen et al., Discrete-Time Observations of Brownian Motion on Lie Groups and Homogeneous Spaces: Sampling and Metric Estimation, Algorithms 2022
>
> [3] Jensen et al., Mean Estimation on the Diagonal of Product Manifolds, Algorithms 2022

---

> ### Author Response · Authors · 2023-11-23
> **Gentle Reminder - Interactive discussion phase will end in less than 3 hours**
>
> Dear Reviewer 8EUf,
>
> We sincerely appreciate your time and efforts in reviewing our paper. As the interactive discussion phase will end in less than 3 hours, we would like to ask if there are any other things you may want us to clarify. We hope our responses have addressed your questions.
>
> Thank you again for your review and valuable suggestions.
>
> Best regards,
> Authors

---

### Official Review · Reviewer_43Ee · 2023-10-30

**Soundness:** 2 fair
**Presentation:** 2 fair
**Contribution:** 3 good
**Rating:** 6
**Confidence:** 5

**Summary:**

This work introduces Riemannian Diffusion Mixture models which aim to build generative models on Riemannian manifolds. Toward this goal, the paper builds SDEs on manifolds that simulate Brownian bridges between two points on the manifold. For simple manifolds, the paper proposes a Logarithmic bridge process conditioned on the target endpoint and uses a noisy geodesic random walk approach to build an SDE. For more general Riemannian manifolds the authors propose Spectral bridge processes that use the eigenvalues associated with the Laplace-Beltrami operator on the manifold to move in a direction that minimizes this spectral distance. Empirically, the Riemannian Diffusion Mixture approach is tested on standard benchmarks for Sphere, Torii, and mesh datasets.

**Strengths:**

There are a few aspects to this paper that are worth highlighting. First learning generic brownian bridge processes on Riemannian manifolds is of special interest to the community and this paper's contribution to this line of work is appreciated. Second, the approach taken in this paper is also quite natural as it leverages two concepts in building a noisy path. 1.) It uses the tangent vector that minimizes the geodesic on the manifold (Logarithmic Bridge process) 2.) It uses the tangent vector that minimizes the spectral distance of the manifold. These distance-minimizing maps are conditioned on an endpoint and provide a simple target. The second idea is an extension of Riemannian Flow Matching (Chen and Lipman 2023) to Brownian bridges (SDEs) and as such quite elegant.

The paper also introduces the idea of two-way bridge matching to reduce the computation of the calculation. Finally, the training objective is divergence-free and can be time-rescaled to minimize variance which aids more numerically stable and accurate training.

**Weaknesses:**

There are several weaknesses in the current draft that I hope the authors can address.

**Presentation**

In a few places, I believe there to be imprecise statements:
- The authors write "the time reversal of the noising process that accompanies score function in the drift which does not
explicitly take into account the underlying geometry.`" --- I don't think this is true as the score function on $\mathcal{M}$ uses the $\textit{Riemannian}$ gradient which needs information on the metric $g$.
- Background: "compact manifolds". Actually, I think you consider, compact, connected, orientable Riemannian manifolds. Otherwise, you cannot define a volume form needed for the integration of probability distributions. Furthermore, the inverse of the expmap may not be well defined. Also, hyperbolic spaces do not fit this definition.
- You haven't introduced the Bolded notation and capitalized notation in eqn 1.
- What is $U$ in equation 1?
- The authors write: "the inverse of the exponential map on the manifold," Not True! For example when the exp map fails to be a diffeomorphism ---i.e. at the cut locus. As a concrete example, in the Lie group, the image of the exponential map of the connected but non-compact group $SL2(R)$ is not the whole group.

**Technical limitations**
I also have several technical concerns that I hope the authors can address.
- The authors write the Bridge process is conditioned on the endpoint $z$. But to build a Brownian bridge you need to pin both endpoints at $t=0$ and $t=T$, otherwise, it's not a bridge. Certainly, that's what is being done when you do two-way bridge matching so this focus and imprecision in the earlier parts of the text are confusing.
- One of the main claims of this paper is that the proposed approach is a lot cheaper computationally because 1.) you do not have to simulate both directions of the SDE and 2.) there is no divergence computation. However, I believe the first part of the claim is a bit misleading because you are effectively trading of variance of the objective for simulation by reusing the same sample in forward and backward matching losses. Moreover, I believe this is exactly why the proposed approach has high variance because to get a low variance estimate you would need to simulate the SDE multiple times as it is noisy so reusing samples here is counter-productive.
- It seems that when you simulate the SDE for training you are effectively caching the training samples. This approach could also readily be applied to Riemannian Flow Matching for general geometries. That is simulates the ODE and caches the samples instead of resampling time uniformly and then simulating the ODE up to that time point for every Loss computation step. So I would love to have a fair comparison to see the numerical benefits to substantiate the faster training claim.
- The authors comment that the time-rescaled proposal distribution $q(t)$ is different than Huang et. al 2022. I disagree, this is the exact same idea that you can resample time instead of uniformly sampling it. Huang et. al 2022 do this using a 1-d flow which is quite cheap and can be done as a pre-processing step while here it is a constant. These are essentially the same but the proposed approach is a cruder approximation.
- The Table 1 results are problematic for two reasons. First, these earth science datasets have known issues and lead to wildly different results based on the subset used for training and test sets. This largely means that we cannot take table numbers from other papers. A quick check shows that the baseline numbers are copied from the papers and looking at the provided code it seems the baselines were not implemented. This leads me to believe that the authors did not try the baselines themselves on the **same training and test splits** as their own reported models. As a result, this table is not useful and cannot be compared. I actually the proposed method will likely be even better if all baselines are compared properly. So I encourage the authors to include this in the updated paper.
- A major claim in the paper is the ability of the method to start from any prior distribution on the manifold to hit any target. This is a strength that vanilla diffusion models do not have. However, I could not find an experiment in the main paper where this is exploited.
- Another claim in the paper is that the method can scale to high dimensional problems, but the largest dimensionality problem is a Torii of $10^3$ dimension. This is hardly a high dimensional manifold as Torii can be written as a product group $S^1 \times S^1 \times \dots \times S^1$ and each manifold in the product is simple and does not need to actually utilize the difficulty that comes with high dimensions. I would consider toning down this claim.


If all of my concerns are addressed I am willing to upgrade my score.

**Questions:**

1.) The noisy paths learned by this approach do not appear to be optimal in the sense of matching the empirical marginal densities. Can we make this better and perhaps more numerically stable by using Optimal transport paths like done in OT-Flow matching (Tong et. al 2023) for Euclidean geometries?

---

> ### Author Response · Authors · 2023-11-16
> **Initial Response to Reviewer 43Ee (1/3)**
>
> We sincerely thank you for your detailed, constructive, and helpful comments. We appreciate your positive comments that
> - Learning bridge processes on Riemannian manifolds are of special interest to the community and this paper's contribution to this line of work is appreciated.
> - The approach taken in this paper is quite natural as it leverages two concepts in building a noisy path. 1.) tangent vector minimizing the geodesic 2.) tangent vector minimizing the spectral distance.
> - Minimizing the spectral distance is an extension of Riemannian Flow Matching to Brownian bridges (SDEs) and as such quite elegant.
> - The paper introduces the idea of two-way bridge matching to reduce the computational cost.
> - Training objective is divergence-free and can be time-rescaled to minimize variance which aids more numerically stable and accurate training.
>
> We provide an updated revision of the paper reflecting your comments (highlighted in orange) and we address all your concerns below.
>
>
> ---
>
> **Comment 1:**
> I don’t think that “score function does not explicitly take into account the underlying geometry” is true.
>
> **Response 1:**
> The purpose of the statement is to explain that the **score function (Riemannian gradient) does not provide an explicit geometrical interpretation** for the diffusion of previous methods. In contrast, for our approach, the drift of the Logarithm bridge yields an **intuitive geometrical explanation** that corresponds to a tangent vector minimizing the geodesic distance toward an endpoint.
>
> We have clarified this point by revising the statement into “... noising process that accompanies score function in the drift which does not provide explicit geometric interpretation.” in the revised paper.
>
> ---
>
> **Comment 2-1:**
> I think you consider, compact, connected, orientable Riemannian manifolds.
>
> **Response 2-1:**
> Thank you for the comment, we do consider **complete, orientable, connected, and boundaryless** Riemannian manifolds, following the assumption of RSGM. We have clarified this point in the revised paper.
>
> ---
>
> **Comment 2-2:**
> Hyperbolic space is not a compact manifold.
>
> **Response 2-2:**
> It appears there was a misunderstanding, as the purpose of the experiment on the hyperbolic space (Section 5.5) is to **empirically show that our framework can be extended to non-compact manifolds**. We demonstrate in Figure 9 that our framework can learn the distribution accurately.
>
> ---
>
> **Comment 3:**
> What is $U$ in Eq. (1)?
>
> **Response 3:**
> $U$ denotes the potential function. For example, $U(x) = d_g(x, \\mu)^2 + \\log \\|D \\exp^{-1}_{\\mu}(x)\\|$ ensures the stationary distribution to be the wrapped Gaussian distribution with an arbitrary mean location $\\mu\\in\\mathcal{M}$, where $d_g$ denotes the geodesic distance. We have added this explanation to the revised version of the paper.
>
> ---
>
> **Comment 4:**
> Exponential map may fail to be a diffeomorphism.
>
> **Response 4:**
> You are right, the exponential map is not injective at the cut locus. Since we are assuming a connected and complete Riemannian manifold, we have clarified in the updated version of the paper that we assume the endpoint is not in the cut locus of the current state.
>
> ---
>
> **Comment 5:**
> Brownian bridge needs to pin both endpoints at $t=0$ and $t=T$.
>
> **Response 5:**
> Although it is true that we need to fix both the starting point ($t=0$) and the endpoint ($t=T$) to define a bridge process, we have omitted to mention the starting point since **the drift of the bridge process solely depends on the endpoint**, describing the bridge process concisely as a “diffusion process conditioned on the endpoint”. We agree that denoting the fixed starting point would make it clearer, and have clarified this in the updated version of the paper.

---

> ### Author Response · Authors · 2023-11-16
> **Initial Response to Reviewer 43Ee (2/3)**
>
> **Comment 6-1:**
> The proposed approach is a lot cheaper computationally because 1.) you do not have to simulate both directions of the SDE and 2.) there is no divergence computation
>
> **Response 6-1:**
> We would like to clarify that apart from these two reasons, **our two-way approach also contributes to greatly reducing the computational cost** during training, since it requires significantly fewer in-training simulation steps compared to the one-way approach, achieving $\\times$34.9 speed up.
>
> ---
>
> **Comment 6-2:**
> You are effectively trading of variance of the objective for simulation by reusing the same sample in forward and backward matching losses.
>
> **Response 6-2:**
> Our method **does not trade the variance using the two-way approach** (i.e., reusing the same sample in forward and backward matching). Reusing the same sample does not result in higher variance since we simulate the bridge processes with different starting and end points sufficiently many times.
>
> We empirically show in the table below that we **obtain almost similar results and variances when using different samples for both forward and backward matching losses**. Further, we would like to clarify that our method does not have high variance compared to the baselines as shown in Tables 1, 2, and Figure 2.
>
> |                                                        | Earthquake                    | Proline                       |
> |--------------------------------------------------------|-------------------------------|-------------------------------|
> | Ours w/ different sample for forward/backward matching | -0.29 $\\pm$ 0.08 | 0.14 $\\pm$ 0.025 |
> | Ours w/ same sample for forward/backward matching      | -0.30 $\\pm$ 0.06 | 0.14 $\\pm$ 0.027 |
>
>
> ---
>
> **Comment 7:**
> When you simulate the SDE for training you are effectively caching the training samples.
>
> **Response 7:**
> We **do not cache the trajectory of the bridge processes during training**. We train the models with the objective in Eq. (12) where we first sample $t\\sim q$ and $(x,y)\\sim (\\Pi,\\Gamma)$ and then simulate the bridge process $\\mathbb{Q}^{x,y}$ in the two-way approach to compute $Z_t$. Note that we do not save any intermediate values for computing the loss during training. We believe that the separate expectations over $Z$ and $t$ caused the misunderstanding and have clarified them in the revised paper.
>
> ---
>
> **Comment 8:**
> The idea of the time-scaled proposal distribution is the same as RDM (Huang et al., 2022).
>
> **Response 8:**
> The **proposal distribution used for the importance sampling in our framework and the one used in RDM are not the same**. We leverage $q(t)\\propto\\sigma^{-2}_t$ as a proposal distribution that is tractable and $t\\sim q$ can be sampled easily with ignorable cost.
>
> On the other hand, RDM uses an intractable proposal distribution and relies on an approximation via an additional neural network that requires separate training. Especially, the training of the importance sampler happens in between the training of the diffusion model $a$, which is not done as a pre-processing step or not done in parallel, and therefore results in extra computational cost and an increase in the training time of the framework.
>
> We would like to emphasize that our approach is much simpler compared to that of RDM while effective in stabilizing the training and improving the generation quality, without the need for additional computation or training time.
>
> We have clarified this point in the revised paper as “While the idea of the importance sampling for the time distribution was also used in Huang et al. (2022), our approach leverages a simple and easy-to-sample proposal distribution $q$, which is effective in stabilizing the training and improving the generation quality, without the need for additional computation or training time.”

---

> > ### Comment · Reviewer_43Ee · 2023-11-19
> > **Re:Response**
> >
> > Hi,
> > Thank you for your response. I'm still working my way through this but in the interest of time. Can you please try to do Fig 10 in your supplementary with 2-Wasserstein distance rather than MMD? I think this is a better metric for SDEs.
> >
> > I will have more to say before the end of the rebuttal period.

---

> > > ### Author Response · Authors · 2023-11-19
> > > **Response to Reviewer 43Ee - Using 2-Wasserstein distance**
> > >
> > > We thank the reviewer for reviewing our paper. We are willing to address any further inquiries you may have.
> > >
> > > ---
> > >
> > > **Comment:** Can you please try to do Fig 10 in your supplementary with 2-Wasserstein distance rather than MMD? I think this is a better metric for SDEs.
> > >
> > > **Response:**
> > > We provide the results using the 2-Wasserstein distance in Figure 11 of the updated version of the paper. From Figure 11, we observe similar results when using MMD:
> > >
> > > - (a) We can obtain sufficiently accurate trajectories with 15 steps for the in-training simulation of our two-way approach.
> > > - (b) One-way approach requires a large number of steps to obtain accurate trajectories.
> > > - (c) 15 steps for the in-training simulation of our two-way approach yield accurate trajectories for our spectral bridge processes.
> > > - (d) When using a small noise scale for our spectral bridge processes, 15 steps produce highly inaccurate trajectories,  which explains the reason for the failure of RFM on mesh datasets, as in Figures 3 and 8.
> > >
> > > In conclusion, the analysis of Section 5.5 remains the same when using the 2-Wasserstein distance instead of MMD: Our method can be trained using only 15 in-training simulation steps.
> > >
> > > ---

---

> > ### Comment · Reviewer_43Ee · 2023-11-19
> > **Re:Response**
> >
> > Thank you for your answers. It seems I misunderstood certain aspects of 2-way bridge matching. I am still confused why doing two-way bridge matching leads to a $\times 34.9$ speedup during training. Can you give a bit more explanation for this? I believe this is a more interesting point that is not sufficiently developed.

---

> ### Author Response · Authors · 2023-11-16
> **Initial Response to Reviewer 43Ee (3/3)**
>
> **Comment 9:**
> Earth science datasets lead to different results based on training and test sets
>
> **Response 9:**
> We have **used the same data split with the reproducible works, RSGM and RFM**, by using the same seed values of 0-4 for the data split. Unfortunately, **the results from some previous works (CNFM, RDM) were not reproducible** as they do not provide open-sourced codes and do not provide details of the data split, and we had no choice but to take the numbers from their paper. Note that we obtained similar numbers to the reported results for RSGM and RFM when running their open-source codes. As the reviewer acknowledges, we believe that our method will likely be even better if all baselines are compared properly, and we are happy to update the results if the codes for other baselines are open-sourced.
>
> ---
>
> **Comment 10:**
> I could not find an experiment in the main paper where the ability of the proposed method to start from any prior distribution on the manifold to hit any target.
>
> **Response 10:**
> By the construction of the Riemannian diffusion mixture (Eq. (6)), it is **theoretically guaranteed that we can use any prior distribution to build a diffusion process to hit the given target distribution**. However, in order to fairly compare with the previous methods that are limited to trivial prior distributions, we have used the same prior distributions with the baseline methods during the experiments. As finding a prior distribution to improve the generation quality is not the main goal of our work, we leave it as future work.
>
> ---
>
> **Comment 11:**
> The paper claims that the method can scale to high dimensions, but the largest dimensionality problem is a Tori of dimension 1000.
>
> **Response 11:**
> Our framework can scale to high dimensions (dimension 2000), which is a notable result considering the limited number of model parameters (MLP with 2 hidden layers), while previous methods (Moser Flow, RSGM, and RFM) fail on lower dimensions (~500). We observe that our method produces consistent results (log p) for higher dimensions when using more model parameters, scaling fairly well even to the tori of dimension 10,000.
>
> ---
>
> **Comment 12:**
> As torus can be written as a product group where each manifold is simple, it does not need to actually utilize the difficulty that comes with high dimensions.
>
> **Response 12:**
> Although it is true that our framework has advantages on high-dimensional tori since the logarithm map can be easily computed due to the fact that tori can be written as a product group, previous works cannot exploit this advantage as the **difficulty of modeling high-dimensional data with previous methods comes from the computation of Riemannian divergence**.
>
> We would like to emphasize that many real-world problems we are interested in consider manifolds with known geodesics, for example, robotics motion planning (SO(3)), protein modeling (SE(3)), neuroscience (SPD matrices), astrophysics (hypersphere),  and computational chemistry (product of tori), and for these problems, our work can provide practical benefits of faster training and reduced computation, as seen in the case of high-dimensional tori.
>
> However, we agree that our framework may also have difficulty when scaling to high-dimensional general manifolds for which computing the logarithm map is expensive. Therefore, we have toned down our claim as “our method can scale to higher dimensions when the geodesics are known.” in the revision.
>
>
> ---
>
> **Comment 13:**
> Can we make the diffusion process of the framework better and perhaps more numerically stable by using Optimal transport paths?
>
> **Response 13:**
>
> We believe one can construct a diffusion process satisfying optimal transport property by leveraging the tools from Schrodinger bridge [1] or stochastic optimal control [2] to the Riemannian setting, which would be a promising direction for future work.
>
> [1] Peluchetti, Diffusion Bridge Mixture Transports, Schrodinger Bridge Problems and Generative Modeling, arXiv 2023
>
> [2] Chen et al., Likelihood Training of Schrodinger Bridge using Forward-Backward SDEs Theory, ICLR 2022
>
> ---
> ---
>
> We thank the reviewer again for their time and feedback, and we hope that our responses have addressed any remaining questions. We hope the reviewer would kindly consider a fresh evaluation of our work given our responses above.

---

> > ### Comment · Reviewer_43Ee · 2023-11-19
> > **Re:Response**
> >
> > Thank you for your responses. I still don't find your argument for comments 11 and 12 convincing. $SO(3)$ is not a high dimensional manifold, neither is $SE(3)$. I believe your framework may work in high dimensional manifolds, but the current empirical evidence does not go beyond Torii so it is not a powerful claim. Having said that I am satisfied with most of your other claims and explanations. Thus I am increasing my score from 3-> 6.

---

> ### Author Response · Authors · 2023-11-20
> **Thank you for your valuable feedback**
>
> We sincerely appreciate the reviewer’s time and effort in reviewing our paper. We are grateful for your valuable feedback that helped enhance the presentation of our paper. We would like to address your comments below:
>
> ---
>
> **Comment 14:**
> $SO(3)$ is not a high dimensional manifold, neither is $SE(3)$.
>
> **Response 14:**
> We apologize for the confusion, we were denoting the high-dimensional Lie groups $SE(3)^{N}$ and $SO(3)^{N}$, for example, the backbone structure of proteins can be represented as an element of the Lie group $SE(3)^{N}$ where $N$ represents the number of residues in the backbone [1, 2, 3].
>
> [1] Jumper et al., Highly accurate protein structure prediction with AlphaFold, Nature 2021
>
> [2] Watson et al., Broadly applicable and accurate protein design by integrating structure prediction networks and diffusion generative models, bioRxiv 2022
>
> [3] Lim et al., SE(3) diffusion model with application to protein backbone generation, ICML 2023
>
> ---
>
> **Comment 15:**
> I believe your framework may work in high dimensional manifolds, but the current empirical evidence does not go beyond Torii so it is not a powerful claim.
>
> **Response 15:**
> We would like to emphasize that the results on tori demonstrate the **advantage of our approach over previous works for high-dimensional manifolds with known geodesics**. While previous works fail for high dimensions due to the computation of Riemannian divergence, our method can scale to higher dimensions for such manifolds since our training objective is divergence-free and the logarithm map can be efficiently computed from the known geodesics.
>
> Following the reviewer’s suggestion, we have toned down our claim as “our framework can scale to high-dimensional manifolds for which the geodesics are known.” in the revised paper.
>
> ---
>
> **Comment 16:**
> I am still confused why doing two-way bridge matching leads to a $\\times$34.9 speedup during training. I believe this is a more interesting point that is not sufficiently developed.
>
> **Response 16:**
> Our two-way approach achieves $\\times$34.9 speedup during training compared to the one-way approach because the **two-way approach requires significantly less number of in-training simulation steps** than the one-way approach.
>
> Specifically, our two-way approach yields accurate trajectories (i.e., $Z_t$ of the bridge process) using only 15 steps for the in-training simulation, while the naive one-way approach requires a large number of simulation steps, at least 300 steps, to obtain accurate trajectories, which we experimentally demonstrate in Figures 10 (a, b) and 11 (a, b).
>
> ---
> ---
>
> Thank you once again for the thoughtful comments that greatly helped improve our paper.
>
> Best,
>
> Authors

---

> > ### Comment · Reviewer_43Ee · 2023-11-20
> > **Re: Comment 16**
> >
> > Yes I understood that it requires less in-training simulation steps. My question was to more towards, why do you suspect that to be the case. Can you give some sort of mathematical argument? Or at the very least some sort of intuition that says more than what you currently have.

---

> ### Author Response · Authors · 2023-11-21
> **Regarding Our Two-way Approach**
>
> Thank you for the question regarding the advantages of our two-way approach.
>
> Unlike the Brownian motion, the drift of the bridge processes dramatically changes for $t$ near the terminal time $T$ due to the $1/(\\tau_T-\\tau_t)$ term, especially when the current state does not match the fixed endpoint. Therefore, when simulating the bridge process in a single direction ($0\\rightarrow t$) with a small number of steps, the error from the discretization is magnified, resulting in inaccurate trajectories which we experimentally demonstrate in Figure 10 (b) (and 11 (b)).
>
> In contrast, our two-way approach does not suffer from this issue because we simulate the reverse of the bridge processes when $t$ is large, for which the norm of the drift of the reverse process is significantly small compared to the forward drifts. Thus the discretization error is dominated by the stochasticity of the processes, and we can accurately compute the trajectories even for large $t$ using a small number of simulation steps.
>
> We appreciate your feedback, and we will add a detailed discussion on the advantages of our two-way approach in the Appendix for the final revision.

---

### Official Review · Reviewer_ZrBb · 2023-10-31

**Soundness:** 3 good
**Presentation:** 3 good
**Contribution:** 2 fair
**Rating:** 6
**Confidence:** 3

**Summary:**

- The paper introduces the Riemannian Diffusion Mixture, a framework for generative modeling on manifolds.
- The proposed framework uses a mixture of endpoint-conditioned diffusion processes to model the generative process on manifolds.
- The paper introduces a simple and efficient training objective that can be applied to general manifolds.
- The method outperforms previous generative models on various manifolds, scales to high dimensions, and requires fewer in-training simulation steps.

**Strengths:**

- The paper addresses the limitations of existing generative models on manifolds, such as expensive divergence computation and reliance on approximations.
- The paper shows the applicability of the framework to general manifolds with non-trivial curvature, including synthetic distributions on triangular meshes and a 2-dimensional hyperboloid.
- Experiment shows better/ comparable results with other methods such as CNFM etc.
- The paper provides a simple and efficient training objective that can be applied to general manifolds.

**Weaknesses:**

- The computation of the Riemannian divergence, which is required for implicit score matching, scales poorly to high-dimensional manifolds. This can limit the applicability of the method to complex datasets.
- The experimental validation of the method may not be comprehensive and may not cover all possible scenarios. The details of the experimental setup and results may not be easy to follow.
- The training time of the proposed framework is not extensively discussed or compared with other methods. While it is mentioned that the proposed method allows for faster training compared to previous diffusion models, a more detailed analysis of the computational efficiency and scalability of the framework would be beneficial.

**Questions:**

- Does the author try such method on higher dimensional manifolds?

---

> ### Author Response · Authors · 2023-11-16
> **Initial Response to Reviewer ZrbB**
>
> We sincerely thank you for your constructive and helpful comments. We appreciate your positive comments that
> - Our work addresses the limitations of previous works: expensive divergence computation and reliance on approximations.
> - Our framework shows the applicability of the framework to general manifolds with non-trivial curvature.
> - Ours achieve better/ comparable results with previous works.
> - Our work provides a simple and efficient training objective that can be applied to general manifolds.
>
> We provide an updated revision of the paper (highlighted in orange), and we initially address all your concerns below.
>
> ---
>
> **Comment 1:**
> Computation of the Riemannian divergence can limit the applicability of the method.
>
> **Response 1:**
> This is a misunderstanding as **our framework does not require computing the Riemannian divergence**. In fact, previous works rely on the Riemannian divergence and thus scale poorly to high dimensions, which we experimentally show in Figure 2 (Right). We emphasize that our method is divergence-free and achieves significantly faster training compared to previous works.
>
> ---
>
> **Comment 2:**
> The experimental validation of the method may not be comprehensive and may not cover all possible scenarios.
>
> **Response 2:**
> We respectfully disagree, as we conduct **extensive experiments in diverse settings on both real-world and synthetic datasets**, including various manifolds (sphere, tori, manifold with non-trivial curvature, hyperbolic space) and different dimensions, to validate the effectiveness of our method.
>
> ---
>
> **Comment 3:**
> The details of the experimental setup and results may not be easy to follow.
>
>
> **Response 3:**
> We believe we have tried our best to describe the details of the implementation details for every experiment, including the baseline, dataset, data split, training method, and model architecture, and further provided an explanation and analysis of all the results. We would greatly appreciate it if the reviewer could specify which part of the explanation for the experimental setup or the result was insufficient.
>
> ---
>
> **Comment 4:**
> The training time of the proposed framework is not extensively discussed or compared with other methods.
>
> **Response 4:**
> We have provided a **comparison of the training time with the baselines in Table 2 (Right)** and discussed in Sections 5.3 and 5.4. Our method achieves up to $\\times$5 speed up for training compared to RSGM based on the Riemannian divergence and further achieves $\\times$12.8 speedup compared to RFM on mesh datasets.
>
> ---
>
> **Comment 5:**
> Does the author try such method on higher dimensional manifolds?
>
> **Response 5:**
> Yes, we show that our method can scale to high dimensions through **experiments on high-dimensional tori (Section 5.3)**.  Figure 2 (Right) demonstrates that our method can scale fairly well even for high dimensions, while previous works fail as the dimension increases showing a significant drop in the performance.
>
> ---
> ---
>
> We thank the reviewer again for their time and feedback. As a last point, we would like to **recap the contributions of our work** briefly:
>
> - We propose a novel diffusion generative framework on manifolds based on a mixture of bridge processes, which is different from the time-reversal approach of previous works.
> - We introduce bridge processes on manifolds and provide an intuitive geometrical interpretation via distance-minimizing tangent vectors.
> - We present a theoretical base for building a mixture of bridge processes on manifolds, which is adaptable to general manifolds and arbitrary prior distributions, unlike the previous diffusion models.
> - We propose a training objective based on the new idea of two-way bridge matching, which is computationally efficient and divergence-free with greatly reduced in-training simulation steps.
> - Our method achieves superior performance on diverse manifolds, that can be trained significantly faster and can scale to higher dimensions compared to previous works.

---

> ### Author Response · Authors · 2023-11-21
> **Gentle Reminder for Reviewer ZrbB**
>
> Dear Reviewer ZrbB,
>
>
> Thank you for reviewing our paper. As the interactive discussion phase will end this Wednesday (22nd AOE), we politely ask you to check our responses. We summarize our responses as follows:
>
>
> - We made clear that our approach does not require the computation of the Riemannian divergence.
> - We clarified that we conducted extensive experiments in diverse settings on both real-world and synthetic datasets.
> - We explained that we have compared the training time with the baselines in Table 2 and discussed in Sections 5.3 and 5.4.
> - We clarified that our method can scale to higher dimensions which we demonstrate in the experiments on high-dimensional tori (Section 5.3).
>
>
> We hope that our responses have addressed your concerns, and we hope you kindly consider updating the rating accordingly. Please let us know if there are any other things that we need to clarify or provide. We sincerely appreciate your valuable suggestions.
>
> Best regards,
> Authors

---

### Author Response · Authors · 2023-11-20
**Global Response by Authors**

Dear Reviewers,

We sincerely thank you for reviewing our paper and for the insightful comments and valuable feedback. We appreciate the positive comments that emphasize the novelty of our work and the advantages of our method:

- Our work **approaches diffusion generative modeling on manifolds with new ideas** (8EUf), which are of **special interest to the community** and the **contribution is appreciated** (43Ee).
- Guiding a diffusion process by **mixing tangent directions is a nice idea** (8EUf) and leveraging the tangent vector that minimizes distance is quite natural and elegant.
- Our work provides a **simple and efficient training objective** (ZrbB, 43Ee).
- Our framework shows **applicability to general manifolds** (ZrbB).

Following the detailed and constructive comments from the reviewers, we have enhanced the presentation by adding the following in the revised paper:

- We modified an unclear sentence in the Introduction regarding the score function as “it does not provide an explicit geometrical interpretation”.
- We specified the assumption on the Riemannian manifolds as complete, orientable, connected, and boundaryless in the Background section.
- We clarified that the inverse of the exponential map is defined out of the cut locus.
- We made clear the starting points of the Brownian bridges in Section 3.
- We clarified the difference in the proposal distribution compared to Huang et al., 2022.
- We have toned down our claim as “our method can scale to higher dimensional manifolds with known geodesics.”

Thank you again for your thorough review and thoughtful suggestions. We hope our responses and the clarifications have addressed any remaining questions, and we are willing to address any further inquiries you may have.

Yours sincerely,

Authors

---

### Meta-Review · Area_Chair_1mxY · 2023-12-08

**Metareview:**

The paper proposes to approach generative modelling on Riemannian manifolds by constructing a mixture of diffusion processes, called Riemannian Diffusion Mixture. The approach is claimed to be scalable to high-dimensional manifolds, compared with other competing methods.

Reviewers generally appreciate that the proposed method for generative modelling on manifolds is new and interesting. However there are some concerns with the current manuscript, especially regarding claims of scalability to high-dimension

- Reviewer 8EUf pointed out that the supposedly new Logarithm Bridge Process is actually a special case of the Fermi bridge (the author put a footnote in the revised version acknowledging this).

- Reviewer 8EUf remains unconvinced about the scalability of the approach to high-dimensional manifolds, unless the Riemamnian logarithm map can be efficiently computed, e.g. has closed form, as later acknowledged by the authors. The same concern for the Spectral Bridge Process, since the eigenvalues and eigenfunctions for the Laplace-Beltrami operator are non-trivial to compute, especially in high-dimension.

- Note that the Spectral Bridge Process would not be applicable to noncompact manifolds such as hyperbolic spaces.

- Reviewer 43Ee pointed out that the only high-dimensional manifold being tested is the tori. Table 1 reports many experimental results, but they are on the 2-D sphere. Since the main claim of the paper is the scalability of the proposed method, more substantial experiments with other high-dimensional manifolds should be given.

**Justification For Why Not Higher Score:**

The experimental validation is not sufficient to support the stated claims of superiority of the proposed method for high-dimensional manifolds.

**Justification For Why Not Lower Score:**

N/A

---

### Decision · Program_Chairs · 2024-01-16

Reject